# Apelin signaling acts as a molecular switch between endothelial and hematopoietic stem cell fates

Jean Eberlein [1], Nadja Groos[1], Navina Shrestha Duwal[1], Wade W Sugden [2,3,4], Trista E North[2,3] & Christian S M Helker [1✉]

## Abstract

Hematopoietic stem and progenitor cells (HSPCs) emerge from arterial endothelial cells (ECs) through a process termed endothelial-to-hematopoietic-transition (EHT), a process induced by paracrine signals and driven by a transcriptional cascade. Despite inductive signals being broadly received by ECs in the dorsal aorta (DA), only a subset of ECs undergoes EHT, while others maintain their vascular identity. The molecular mechanisms that determine this selective fate decision remain poorly understood. Here, we discover Apelin signaling as a critical regulator of cell fates in the DA, acting as a molecular switch to balance vascular and hematopoietic identities. We show that Apelin receptor (Aplnr)-expressing ECs retain their arterial identity, while Aplnr non-expressing ECs are primed to become hemogenic endothelial cells (HECs) and transition into HSPCs. Loss of Apelin signaling leads to excessive EC-to-HEC conversion and increased HSPC numbers. Conversely, forced Aplnr expression abolishes HSPC formation by maintaining EC identity. These findings reveal that Apelin signaling regulates HSPC formation by preserving endothelial identity. In summary, our findings establish Apelin signaling as a critical regulator for balancing endothelial and hematopoietic fates.

**Keywords** Hemogenic Endothelium; Endothelial-to-hematopoietic Transition; Hematopoietic Stem Cells; Zebrafish; Apelin Signaling
**Subject Categories** Signal Transduction; Stem Cells & Regenerative Medicine; Vascular Biology & Angiogenesis

## Introduction

The process of hematopoiesis is highly conserved among vertebrates and occurs in two major waves (Thompson et al, 1998; Galloway and Zon, 2003). The primitive wave gives rise to transient populations of erythroid and myeloid precursor cells (Palis et al, 1999; Herbomel et al, 1999; Warga et al, 2009). During the

definitive wave hemogenic endothelial cells (HECs) in the aorta-gonad-mesonephros (AGM)/ventral wall of the dorsal aorta (VDA) differentiate into multipotent hematopoietic stem and progenitor cells (HSPCs), in a process termed endothelial-to-hematopoietic transition (EHT) (Bertrand et al, 2010a; Kissa and Herbomel, 2010; Boisset et al, 2010; Bonkhofer et al, 2019; Monteiro et al, 2016). A subset of these HSPCs is long-lived and multi-potent, with the ability to self-renew and differentiate to maintain all mature blood cells.

The transcription factors Gata2 and Runx1 are essential for the development of definitive HSPCs across vertebrates (Lam et al, 2010; North et al, 2002; Dobrzycki et al, 2020; Gioacchino et al, 2021; Butko et al, 2015; Kalev-Zylinska et al, 2002). In zebrafish, Gata2b acts upstream of Runx1, the earliest reported marker for functional HSPCs, and is detectable in the hemogenic endothelium as early as 20 hpf (Butko et al, 2015). However, progenitors that have the potential to give rise to HSPCs are specified even earlier, prior to *gata2b* expression. During the initial phase, angioblasts in the lateral plate mesoderm acquire hematopoietic potential through a cascade involving Wnt and Notch signaling as they migrate toward the midline (Clements et al, 2011; Lee et al, 2014). Furthermore, BMP4 secreted from the ventral mesenchyme underlying the DA, induces *runx1* expression in the VDA (Wilkinson et al, 2009). Once HECs are specified and committed toward HSPC fate via *runx1*, they start to express *myb* (Murayama et al, 2006) before expressing *itga2b* later in development (Bertrand et al, 2008; Kissa et al, 2008). These nascent HSPCs have the potential to develop into erythroid, lymphoid and myeloid lineages, although a number of recent studies suggest that HSPCs derived from DA/AGM are a heterogenous population with specific blood lineage biases and that this heterogeneity is inherited from the (hemogenic) endothelium prior to EHT (Xia et al, 2023; Ghersi et al, 2023; Kasper et al, 2020). Following EHT, HSPCs bud out from the DA, enter circulation and colonize a secondary vascular stem cell niche, the caudal hematopoietic tissue (CHT), presumed to be the zebrafish equivalent to the mammalian fetal liver (Murayama et al, 2006; Tamplin et al, 2015). Within the CHT, HSPCs can rapidly expand and begin to differentiate (Xue et al, 2017; Jin et al, 2007; Tamplin et al, 2015). Finally, HSPCs leave this transient niche and populate the kidney, the zebrafish equivalent to the mammalian bone marrow (Murayama et al, 2006; Lam et al, 2010; Zapata, 1979). Although only a small pool of HSCs is formed during embryonic

[1]Department of Biology, Animal Cell Biology, Marburg University, Marburg, Germany. [2]Stem Cell Program, Division of Pediatric Hematology and Oncology, Boston Children's Hospital, Boston, MA 02115, USA. [3]Harvard Stem Cell Institute, Cambridge, MA 02138, USA. [4]Present address: Department of Cell Biology, Neurobiology and Anatomy, Versiti Blood Research Institute, Milwaukee, WI 53226, USA. ✉E-mail: christian.helker@biologie.uni-marburg.de

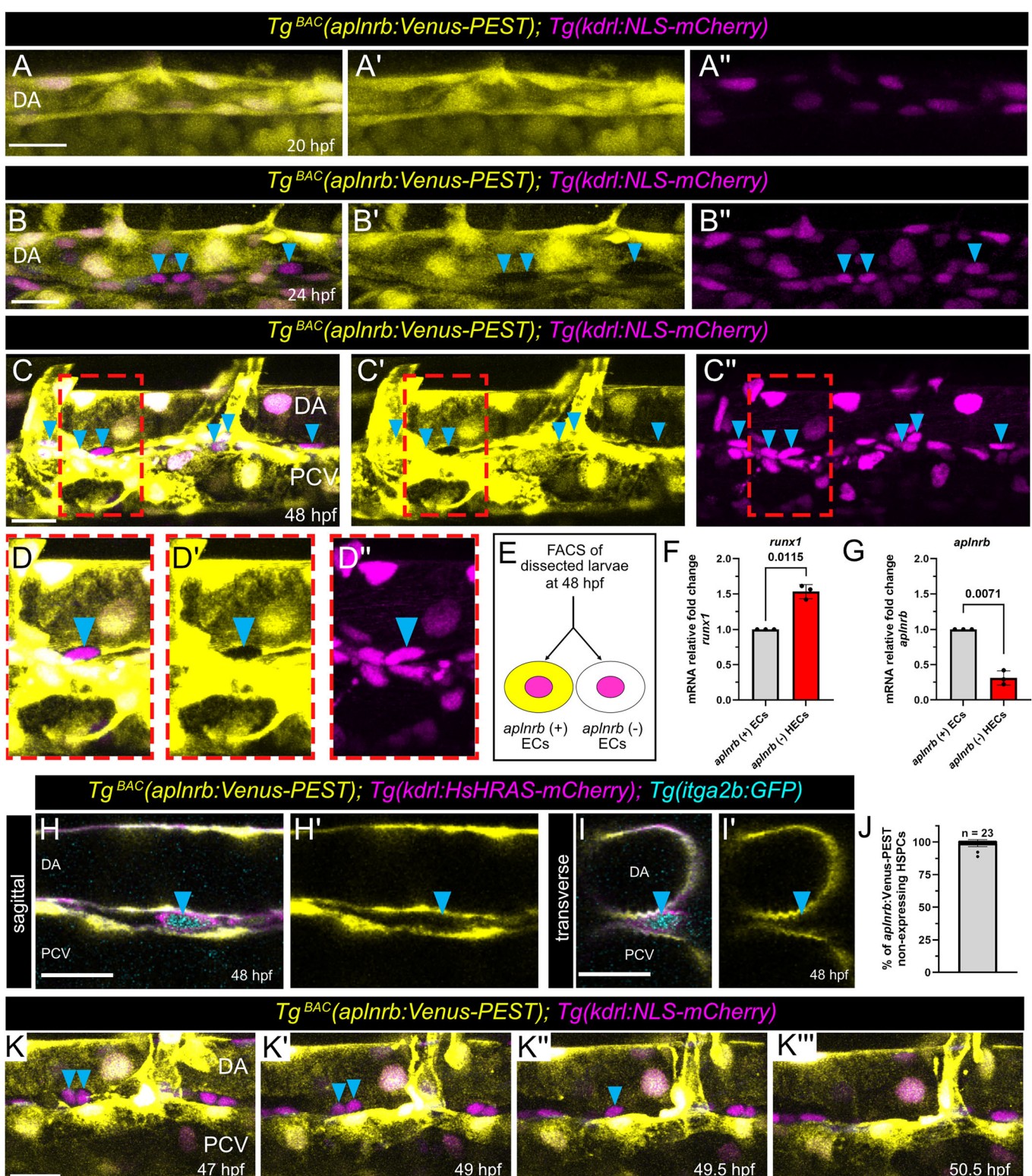

development (Henninger et al, 2017), these HSCs are capable of generating all blood lineages throughout life. Consequently, defects in the development of HSCs during embryonic hematopoiesis can lead to a variety of hematopoietic diseases in adulthood. Apelin signaling via the Apelin receptor (Aplnr), a 7-transmembrane G-protein-coupled receptor (GPCR), plays a crucial role in cardiovascular development (Zeng et al, 2007; Helker et al, 2020, 2015; Scott et al, 2007; Malchow et al, 2024; Herdt et al, 2025; Del Toro et al, 2010; Kidoya et al, 2008). The Aplnr has been shown to couple primarily to the Gαi and Gαq protein families (Shin et al, 2018), initiating downstream signaling

**Figure 1.  *aplnrb* non-expressing endothelial cells give rise to HSPCs.**

(A–D) Confocal projection images of the trunk region of *Tg^BAC^(aplnrb:Venus-PEST); Tg(kdrl:NLS-mCherry)* double transgenic zebrafish embryos at 20 hpf (A–A") and 24 hpf (B–B") and 48 hpf (C–D"). (B–D") *aplnrb*:Venus-PEST non-expressing ECs can be detected in the ventral DA (cyan arrowheads) at 24 (B) and 48 hpf (C, D). (D–D") Magnification of the indicated region in (C). (E) Schema displaying the FACS modalities for quantitative RT-PCR sample generation. (F, G) Quantitative RT-PCR results for *runx1* (paired *t* test, P = 0.0115) and *aplnrb* (paired *t* test, P = 0.0071) on FACS sorted *aplnrb*:Venus-PEST expressing and non-expressing ECs. (H–I') Sagittal and transverse slices of a confocal projection image of a triple transgenic *Tg^BAC^(aplnrb:Venus-PEST); Tg(kdrl:HsHRAS-mCherry); Tg(itga2b:GFP)* zebrafish embryo at 48 hpf. Sagittal (H) and transverse (I) slice showing a single *aplnrb*:Venus-PEST negative HEC expressing *itga2b*:GFP and *kdrl*:HsHRAS-mCherry (cyan arrowhead). (J) Percentage of *aplnrb*:Venus-PEST non-expressing HSPCs in the VDA at 48 hpf, corresponding to (H–I') (mean = 99,1825%, n = 23 embryos). (K–K"') Still images from confocal time-lapse movie of a double transgenic *Tg^BAC^(aplnrb:Venus-PEST); Tg(kdrl:NLS-mCherry)* zebrafish embryo. *aplnrb*:Venus-PEST negative ECs (arrowheads) bud out of the DA. DA dorsal aorta, PCV posterior cardinal vein. Quantifications are displayed as mean ± SD; Scale bars: 20 μm.

through pathways such as MEK and ERK1/2, PI3K and AKT (Malchow et al, 2024; Masri et al, 2006; Tatemoto et al, 2001). In both mice and zebrafish, Aplnr is highly expressed in newly sprouting angiogenic blood vessels (Kidoya et al, 2008; Helker et al, 2020). In addition, Aplnr expression has been observed in the AGM region in mice (Vink et al, 2020; Jackson et al, 2021) and humans (Crosse et al, 2020; Jackson et al, 2021). However, reports regarding the function of Apelin signaling during developmental hematopoiesis are conflicting and contradictory (Chen et al, 2019; Yu et al, 2012; Jackson et al, 2021). Here, we identify an unknown role for Apelin signaling in maintaining EC identity and thereby restricting the formation of HECs in the vascular compartment. Using live confocal imaging of novel transgenic reporters, we identified a zonation within the DA, characterized by *apelin receptor b (aplnrb)* expressing and non-expressing ECs. Loss of Apelin signaling resulted in excessive EC-to-HEC conversion, leading to an expansion of hemogenic endothelial cells (HECs) and an increase in hematopoietic stem and progenitor cell (HSPC) numbers. Collectively, our data suggest that Apelin signaling serves as a molecular switch, maintaining vascular endothelial identity and restricting the hemogenic endothelium to regulate HSPC formation.

# Results

## apelin receptor b (aplnrb) expression defines two endothelial cell subpopulations in the dorsal aorta

In order to study Apelin signaling during hematopoietic development, we analyzed the expression of the *aplnrb* in the DA. To this end we genetically labeled *aplnrb* expressing cells using the *Tg^BAC^(aplnrb:Venus-PEST)* and the nucleus of endothelial cells with mCherry using the *Tg(kdrl:NLS-mCherry)* zebrafish line. At 20 hpf we observed *aplnrb*:Venus-PEST expression in all ECs within the DA (Fig. 1A). Surprisingly, starting at 22 hpf (Fig. EV1A), and more prominently at 24 hpf (Fig. 1B), we noted a subpopulation of ECs within the VDA which had lost expression of *aplnrb*:Venus-PEST. At 48 hpf, both *aplnrb*:Venus-PEST expressing, and non-expressing ECs were still present in the VDA (Figs. 1C,D and EV1B,B'). However, by 72 hpf, *aplnrb*:Venus-PEST expression was absent from the DA and PCV and restricted to the intersegmental vessels (Fig. EV1C). To exclude labeling of non-EC cells by the *Tg(kdrl:NLS-mCherry)* transgenic line, we analyzed *Tg^BAC^(aplnrb:Venus-PEST); Tg(kdrl:HsHRAS-mCherry)* embryos. In line with our prior analysis, we detected a subset of *aplnrb*:Venus-PEST non-expressing ECs exclusively in the VDA (Fig. EV1D–E"). To this end, we also analyzed *Tg(kdrl:NLS-mCherry); (Tg(kdrl:EGFP)* double transgenic embryos and found a

complete overlap of mCherry and GFP expression (Fig. EV1F–G'). Together this indicated that a subpopulation of ECs in the VDA downregulate *aplnrb*:Venus-PEST expression during development of the DA.

## apelin receptor b (aplnrb) non-expressing endothelial cells are pre-primed hematopoietic stem and progenitor cells

Based on their spatio-temporal location in the VDA, we speculated that *aplnrb*:Venus-PEST non-expressing ECs represent HECs and nascent HSPCs. To determine if the *aplnrb* non-expressing cells are HECs, we assessed the mRNA levels of hematopoietic and endothelial marker genes in the *aplnrb* expressing and non-expressing cell fractions. ECs within the trunk of 48 hpf old zebrafish embryos were isolated by FACS based on *aplnrb*:Venus-PEST expression (Fig. 1E) and utilized for quantitative RT-PCR analysis. qRT-PCR revealed that the *aplnrb* non-expressing ECs in the ventral floor of the DA exhibit elevated mRNA levels of the conserved HEC/HSPC marker gene *runx1* (Fig. 1F), with decreased expression of *aplnrb* mRNA (Fig. 1G). Furthermore, we noted the *aplnrb* non-expressing ECs showed reduced expression of the pan-vascular marker *kdrl (vegfr2)*, suggestive of loss of vascular identity (Fig. EV1H) while acquiring a HEC fate (Fig. 1F). To further verify that the *aplnrb* non-expressing cells in the VDA represent HECs/HSPCs, we performed confocal imaging of the *Tg^BAC^(aplnrb:Venus-PEST)* line with the *Tg(itga2b:GFP)* line, (also known as *Tg(CD41:GFP)*) and observed that *aplnrb* non-expressing cells express the *itga2b*:GFP HSPC reporter (Fig. 1H–I'). Quantitative analysis further revealed that all HSPCs in the VDA originate from *aplnrb* non-expressing HECs (Fig. 1J). Of note, we did not detect expression of *apln*, a ligand of the Aplnr at 24 hpf (Fig. EV1I,I') or at 48 hpf (Fig. EV1J,J') in EC of the DA, but rather in neighbouring tissues. Finally, we investigated the behavior of the *aplnrb* non-expressing cells in the VDA by confocal time-lapse imaging (Fig. 1K,K"'). Notably, from 47 to 50.5 hpf, we observed that *aplnrb* non-expressing cells in the VDA emerge from the DA into the subaortic space, and subsequently enter circulation through the PCV (Fig. 1K–K"'), a process characteristic for HSPC emergence (Kissa et al, 2008). Together, these results indicate that *aplnrb* non-expressing ECs in the DA represent newly specified HECs and nascent HSPCs.

## Apelin signaling balances EC and HSPC fates in the DA

Since we found that *aplnrb* non-expressing ECs give rise to HSPCs, we aimed to further investigate the functional requirement of Apelin signaling during this process. To visualize HECs/HSPCs, we

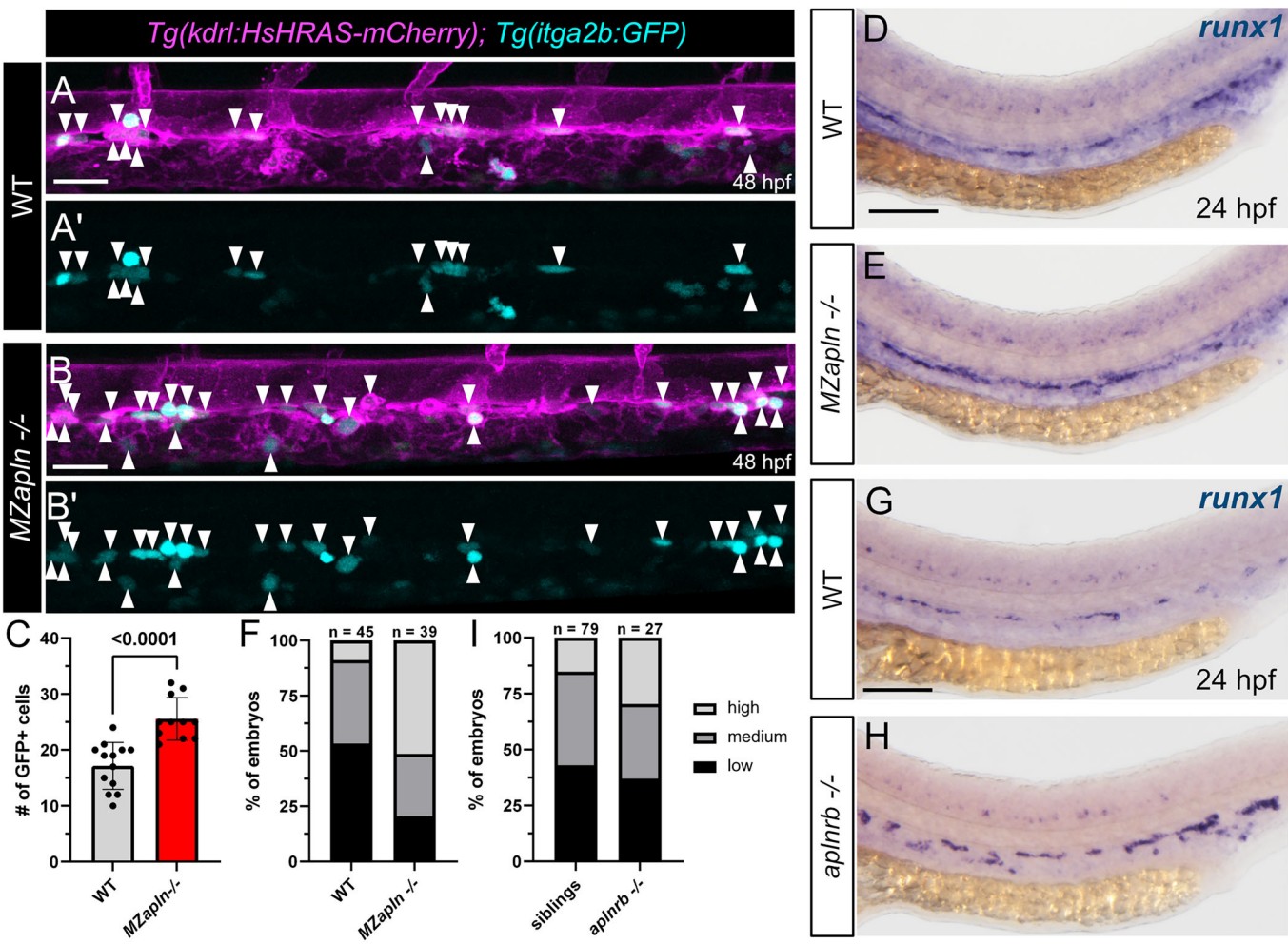

**Figure 2. Elevated HSPC numbers in the VDA upon loss of Apelin signaling.**

(A–B') Confocal projection images of double transgenic *Tg(kdrl:HsHRAS-mCherry); Tg(itga2b:GFP)* zebrafish embryos at 48 hpf. *itga2b*:GFP positive HSPCs in the VDA are highlighted by white arrowheads. (A, A') Trunk region of a wild-type zebrafish embryo. (B, B') Trunk region of a *MZapln* −/− mutant zebrafish embryo. Elevated HSPC numbers compared to wild-type embryos can be observed. (C) Quantification of *itga2b*:GFP positive cells in the DA, corresponding to (A, B) (n = 24, unpaired *t* test, *P* < 0.0001). Quantifications are displayed as mean ± SD. (D, E) WISH for *runx1* at 24 hpf in wild-type and *MZapln* −/− mutant zebrafish embryos, showing increased *runx1* expression in *MZapln* −/− mutant embryos. (F) Phenotypic distribution plot of embryos in (D, E) (n = 84). (G, H) WISH for *runx1* at 24 hpf in siblings and *aplnrb* −/− mutant zebrafish embryos, showing increased *runx1* expression in *aplnrb* −/− mutant embryos. (I) Phenotypic distribution plot of embryos in (G, H) (n = 106). HSPC hematopoietic stem and progenitor cell, VDA ventral dorsal aorta, DA dorsal aorta. Scale bars: 30 µm (A, B); 100 µm (D–H).

performed confocal imaging of the transgenic zebrafish line *Tg(itga2b:GFP)*. In the DA of maternal-zygotic *apln* (*MZapln*) mutants, we observed an increased number of HSPCs in the VDA at 48 hpf (Fig. 2A–C), whereas zygotic *apln* mutants showed no significant change in HSPC numbers (Fig. EV2). We next asked if *aplnrb*, the zebrafish *aplnr* orthologue with the most penetrant vascular phenotypes associated with the mouse/human gene, is responsible for these Apelin-dependent hematopoietic effects. However, homozygous *aplnrb* mutants exhibited a cardiac defect and lack of blood flow (Deshwar et al, 2016; Qi et al, 2022; Zeng et al, 2007; Scott et al, 2007), previously shown to be a key regulator of HSPC maintenance and emergence (Poullet et al, 2019; Lundin et al, 2020; Campinho et al, 2020; Lancino et al, 2018; North et al, 2009). Therefore, we only investigated the effects of loss of *aplnrb* on hematopoiesis before the onset of circulation. In zebrafish, *runx1* expression marks HECs in the DA from 23 hpf onwards

(Wilkinson et al, 2009). Supporting our confocal studies, *runx1* mRNA expression was increased in the VDA in *MZapln* and *aplnrb* mutants at 24 hpf (Fig. 2D–I). Notably, both *MZapln* and *aplnrb* mutants exhibited normal arterial development, as shown by WISH for the arterial markers *flt1* and *ephrin-B2a* at 24 hpf (Fig. EV3A–D'). Furthermore, expression of the pan-endothelial marker *etv2* is also unaltered upon loss of Apelin signaling (Fig. EV3E–F'). Lastly, we investigated a possible role of Apelin signaling in primitive hematopoiesis. We observed no difference in the expression of the erythroid marker *gata1a* or the granulocyte marker *mpx* in *MZapln* and *aplnrb* mutants compared to control embryos (Fig. EV3G–J'). In addition, we analyzed a possible role of Apelin signaling on erythro-myeloid progenitors (EMPs) in the posterior blood island (PBI) by confocal microscopy (Bertrand et al, 2010b). Expression of the transgenic markers *Tg(itga2b:GFP)* and *Tg(gata1:DsRed)* allows to distinguish between HSCs and

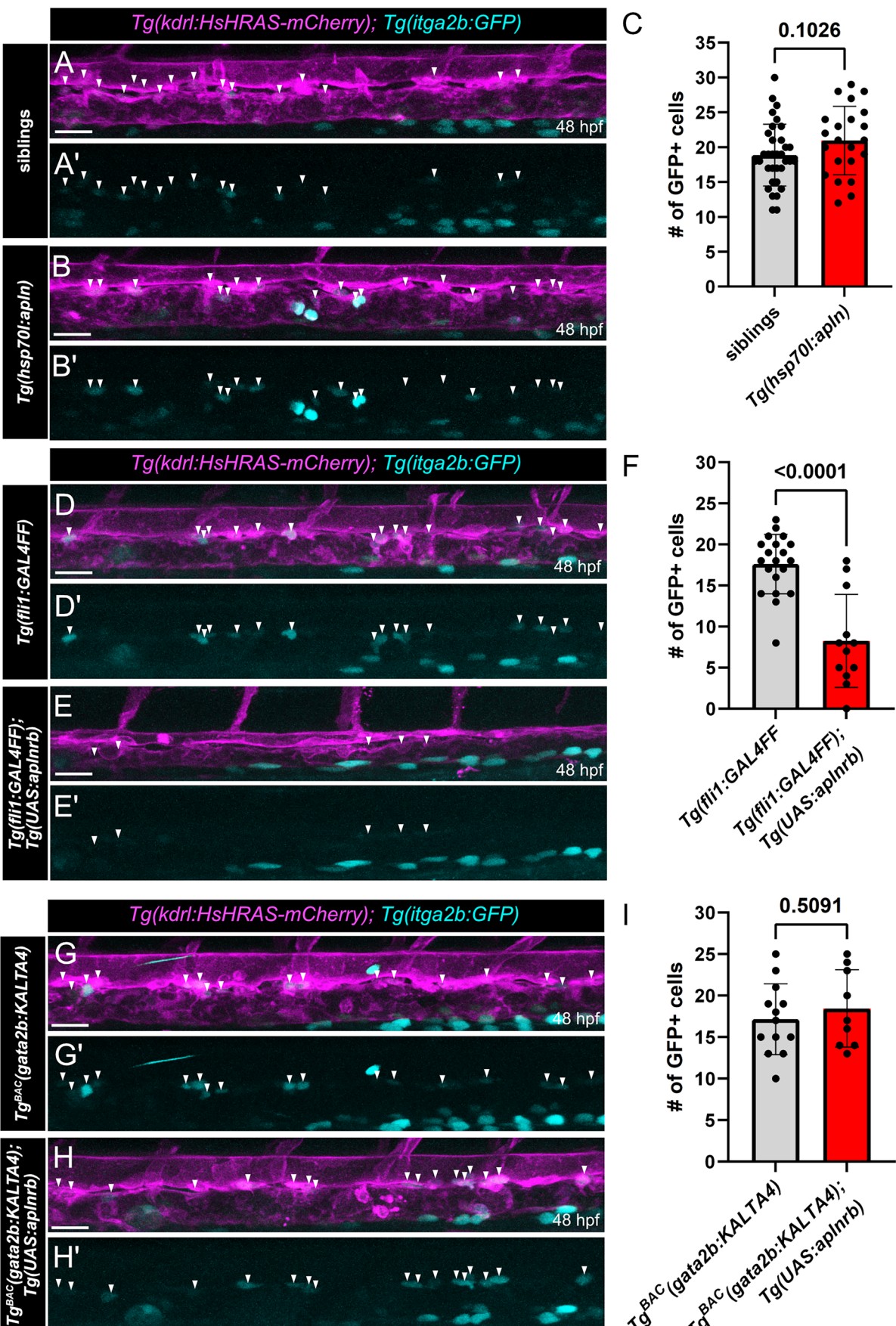

Figure 3.   Forced expression of *aplnrb* in the vasculature inhibits HE specification.

(A, B, D, E, G, H) Confocal projection images of double transgenic *Tg(kdrl:HsHRAS-mCherry); Tg(itga2b:GFP)* zebrafish embryos at 48 hpf. *itga2b*:GFP positive HSPCs are highlighted by white arrowheads. (A–B') Trunk region of transgenic zebrafish embryos expressing *hsp70l*:apln and control siblings, mediating global overexpression of *apln* induced by a heatshock at 24 hpf. No difference in HSPC (white arrowheads) numbers can be observed upon global *apln* overexpression. (C) Quantification of *itga2b*:GFP positive HSPCs in the DA, corresponding to (A, B) ($n = 56$, unpaired $t$ test, $P = 0.1026$). (D–E') Trunk region of transgenic zebrafish embryos expressing *fli1*:GAL4FF (control) or *fli1*:GAL4FF/*UAS:aplnrb*, mediating vascular overexpression of *aplnrb*. A strong reduction in the number of *itga2b*:GFP positive HSPCs (white arrowheads) can be observed upon forced expression of *aplnrb*. (F) Quantification of *itga2b*:GFP positive HSPCs in the DA, corresponding to (D, E) ($n = 32$, unpaired $t$ test, $P < 0.0001$). (G–H') Trunk region of transgenic zebrafish embryos expressing *gata2b*:KALTA4 (control) or *gata2b*:KALTA4/*UAS:aplnrb*, mediating hemogenic endothelium specific overexpression of *aplnrb*. No differences in the number of *itga2b*:GFP positive HSPCs can be observed. (I) Quantification of *itga2b*:GFP positive HSPCs in the DA, corresponding to G and H ($n = 22$, unpaired $t$ test, $P = 0.5091$). HSPCs hematopoietic stem and progenitor cells, DA dorsal aorta. Quantifications are displayed as mean ± SD. Scale bars: 30 µm.

EMPs. EMPs co-express *itga2b*:GFP and *gata1*:DsRed, while HSCs only express *itga2b*:GFP (North et al, 2016). We found that the number of EMP cells in the PBI is not affected by loss of Apelin signaling (Fig. EV4A–C). Altogether, this data suggests that Apelin signaling specifically restricts HEC formation and thereby modulates HSPC development in the DA.

To test if overexpression of the Apelin ligand is sufficient to restrict HSPC development, we used a heat shock-inducible transgenic line (*Tg(hsp70l:apln)*) to ubiquitously overexpress *apln*. The embryos were heat-shocked at 24 hpf and the effect of *apln* overexpression was analyzed by confocal imaging of *Tg(kdrl:HsHRAS-mCherry); Tg(itga2b:GFP)* double transgenic zebrafish embryos at 48 hpf. Overexpression of the ligand *apln* does not influence the number of HSPCs in the VDA at 48 hpf (Fig. 3A–C). We overexpressed *apln* at two further timepoints during HEC specification (Fig. EV5A) and HSPC maintenance (Fig. EV5B) but could not observe changes in the number of HSPCs in the DA at 48 hpf (Fig. EV5). Similarly, global *apln* overexpression did not change the expression of the HEC/HSPC marker *runx1* at 24 hpf (Fig. EV6A–C). This was concordant to our hypothesis, as the HECs/HSPCs in the VDA do not express the receptor *aplnrb* and are therefore unable to react to an overexpression of the ligand Apelin.

To prove our hypothesis that Apelin signaling restricts the hemogenic fate of ECs in the DA, we next overexpressed the receptor *aplnrb* in all ECs. Toward this aim we generated a novel transgenic zebrafish line *Tg(UAS:aplnrb)* which allows us to overexpress *aplnrb* tissue specific by use of the Gal4/UAS system. Overexpression of *aplnrb* in all ECs, mediated by *Tg(fli1:GAL4FF)*, led to a drastic reduction in the number of *itga2b*:GFP expressing HSPCs in the DA at 48 hpf (Fig. 3D–F). We confirmed this data by WISH for the HEC/HSPC marker *runx1* at 24 hpf and could observe the same phenotype (Fig. EV6D–F). In contrast, HEC specific overexpression of *aplnrb*, mediated by *Tg<sup>BAC</sup>(gata2b:KALTA4)*, had no impact on the number of *itga2b*:GFP expressing HSPCs in the DA at 48 hpf (Fig. 3G–I). Similar findings were noted for the expression of *runx1*, which was unaltered upon HEC specific *aplnrb* overexpression at 24 hpf (Fig. EV6G–I). Together, our data indicates that *aplnrb* expression needs to be downregulated before the specification of HECs.

## Elevated HSPC numbers result from enhanced EC-to-HEC transdifferentiation in Apelin-deficient embryos

To investigate the mechanisms underlying the increase in hematopoietic stem and progenitor cell (HSPC) numbers due to altered

Apelin signaling, we first assessed whether heightened proliferation of HSPCs in the ventral dorsal aorta (VDA) could explain this increase. Using EdU labeling assays, we examined proliferative activity by quantifying EdU-positive cells in the VDA of *aplnrb* mutant embryos and their control siblings at 36 h post-fertilization (hpf). Our results revealed no significant difference in the number of proliferating cells between *aplnrb* mutants and control siblings, indicating that increased HSPC numbers in Apelin-deficient embryos are not due to enhanced HSPC proliferation in the VDA (Fig. 4A–C). We then speculated that Apelin signaling may restrict HEC fate in arterial ECs and therefore, loss of Apelin signaling would lead to an expansion of the HEC pool. To test this hypothesis, we employed a genetic approach by injecting three crRNAs targeting different exons (Kroll et al, 2021; Quick et al, 2021) of the *apln* gene into *Tg<sup>BAC</sup>(aplnrb:Venus-PEST); Tg<sup>BAC</sup>(gata2b:KALTA4); Tg(UAS:lifeact-GFP)* embryos and analyzed the F0 CRISPants at 48 hpf (Fig. 4D–G). Embryos injected with crRNAs targeting *apln* exhibited an increased number of HSPCs in the VDA (Fig. 4D–F), consistent with our previous results. Most strikingly, we observed that around 20% of the *gata2b* reporter expressing HECs/HSPCs co-expressed *aplnrb*:Venus-PEST (Fig. 4D–E',G). This suggests that, in the absence of Apelin signaling, uncommitted ECs, which would normally retain their vascular identity, can instead adopt a HEC fate (Fig. 4D–E',G). These findings suggest that the loss of Apelin signaling creates a permissive state within ECs, allowing a broader specification of HECs. Consequently, the increased HSPC numbers in Apelin-deficient embryos result from the enhanced conversion of ECs into HECs.

## *aplnrb* non-expressing HEC progenitors form independent of major regulators of hematopoiesis

To explore the molecular mechanisms underlying the zonation of arterial endothelial cells (ECs) into *aplnrb* expressing and non-expressing populations, we investigated several known regulators of embryonic hematopoiesis. Notch signaling, a critical pathway for HSPC specification (Kim et al, 2014), was assessed for its role in modulating *aplnrb* expression. To test this, we treated *Tg<sup>BAC</sup>(aplnrb:Venus-PEST); Tg(kdrl:NLS-mCherry)* double-transgenic embryos with the γ-secretase inhibitor RO4929097 at specific developmental timepoints (Fig. EV7A–C'). Neither early Notch inhibition during axial migration (Fig. EV7B,B'; treatment from 6 to 15 hpf), nor a later inhibition during HSPC specification in the VDA (Fig. EV7C,C'; treatment from 15 to 40 hpf) lead to an altered pattern of *aplnrb*:Venus-PEST expressing and non-expressing cells in the DA compared to DMSO treated control embryos (Fig. EV7A,A'). BMP4 secreted from the ventral

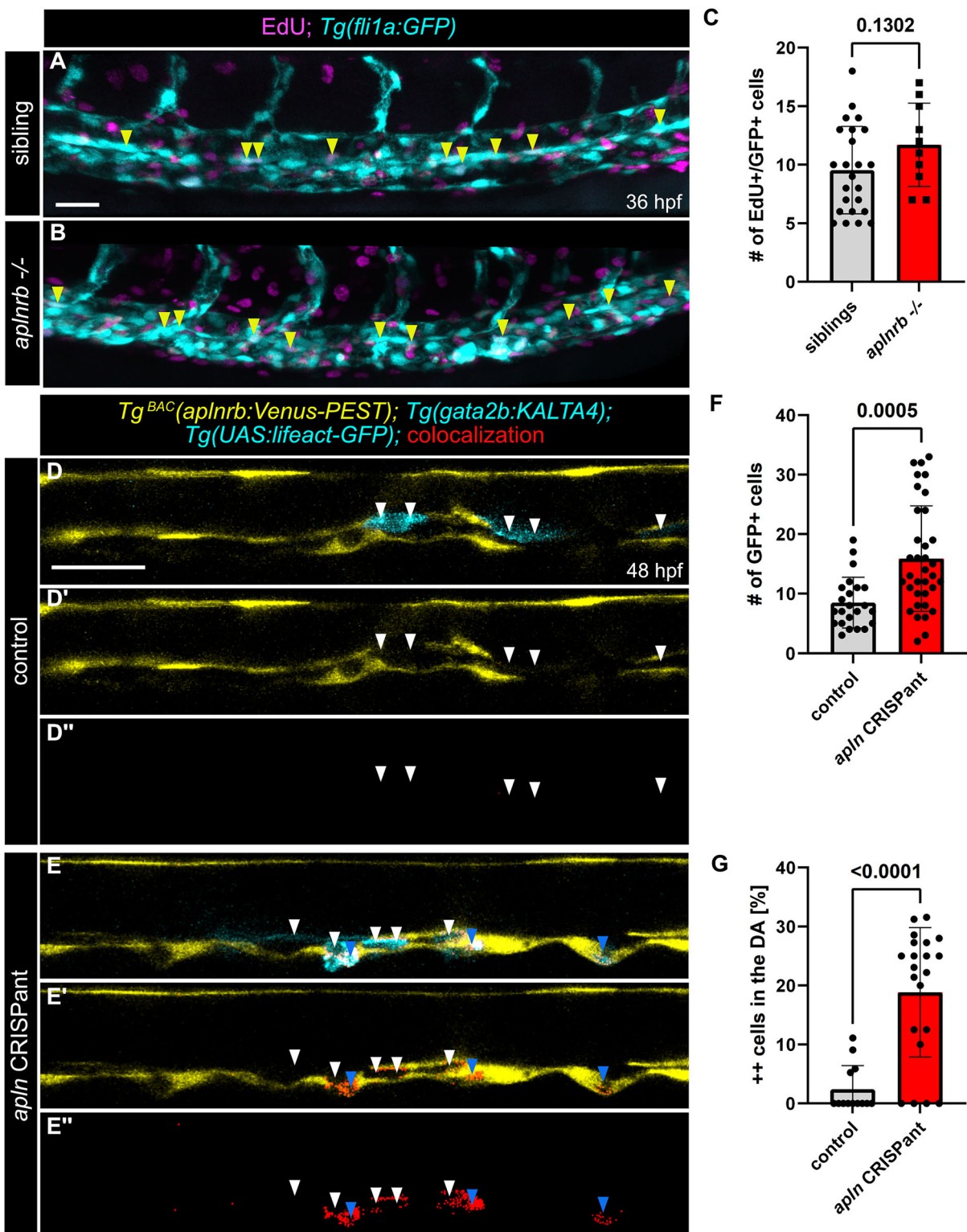

mesenchyme underlying the DA functions to induce *runx1* expression within the VDA (Wilkinson et al, 2009). To determine whether BMP signaling influences *aplnrb* expression in this region, we treated embryos with the BMP inhibitor DMH-1 (10 μM). This treatment did not affect the *aplnrb* expression pattern in the DA (Fig. EV7D,D'). Similarly, given the importance of somitic Wnt16/Notch signaling in HEC specification(Clements et al, 2011; Kim et al, 2014), we assessed the impact of ectopic Wnt activation on *aplnrb* expression. However, no changes in *aplnrb*:Venus-PEST expression were observed following Wnt activation (Fig. EV7E,E').

**Figure 4. Enhanced conversion of ECs-to-HECs in Apelin signaling deficient larvae.**

(A, B) Confocal projection images of Tg(fli1a:GFP) zebrafish embryos at 36 hpf. Proliferating cells (magenta) are labelled by EdU incorporation assay. Yellow arrowheads label proliferating cells in the VDA. (A) Trunk region of a control sibling zebrafish embryo. (B) Trunk region of an aplnrb −/− mutant zebrafish embryo. (C) Quantification of the proliferating cells in the VDA corresponding to (A, B). (n = 34, unpaired t test, P = 0.1302). (D–E') Sagittal slices of confocal projection images of triple transgenic Tg^BAC(aplnrb:Venus-PEST); Tg(gata2b:KALTA4); Tg(UAS:lifeact-GFP) zebrafish embryos at 48 hpf. Colocalization of GFP and Venus signal is displayed in red. (D–D") trunk region of a control zebrafish embryo. aplnrb negative HECs/HSPCs are highlighted by white arrowheads. (E–E") trunk region of an apln CRISPR injected zebrafish embryo. aplnrb negative HECs/HSPCs are highlighted by white arrowheads. aplnrb positive ectopic HECs/HSPCs are highlighted by blue arrowheads. (F) Quantification of GFP+ HSPCs in the DA, corresponding to (D–E'). (n = 58, unpaired t test, P = 0.0005). (G) Quantification of GFP/aplnrb:Venus-Pest double positive cells, displayed as percentage of total GFP+ cells. (n = 34, unpaired t test, P < 0.0001). Quantifications are displayed as mean ± SD. Scale bars: 30 μm.

Furthermore, Hedgehog signaling is known to be involved in HSPC development (Wilkinson et al, 2009; Gering and Patient, 2005). To examine whether Hedgehog signaling influences aplnrb expression, we treated zebrafish embryos with 20 μM Purmorphamine to activate the Smoothened receptor (Sinha and Chen, 2006). Again, ectopic Hedgehog signaling did not appear to alter the aplnrb:Venus-PEST expression pattern in the DA (Fig. EV7F,F'). Finally, the transcription factor runx1 is an essential regulator of the transition from endothelial to hematopoietic fate. Zebrafish mutant for runx1 show impaired hematopoiesis (Kissa and Herbomel, 2010) and fail to downregulate arterial genes in HECs (Bonkhofer et al, 2019). To determine whether Runx1 is required for the downregulation of aplnrb expression in the VDA, we analyzed double transgenic Tg^BAC(aplnrb:Venus-PEST); Tg(kdrl:NLS-mCherry) runx1 mutant embryos. Homozygous runx1 mutants, displayed an unchanged aplnrb:Venus-PEST expression pattern in the DA (Fig. EV7G–H'), suggesting that Runx1 is not required for the spatial restriction of aplnrb expression in the DA. Although loss of Runx1 function did not alter the aplnrb expression pattern in our experiments, previous studies have shown, that loss of runx1 leads to abortive EHT events (Kissa and Herbomel, 2010). To investigate whether the loss of Apelin signaling could rescue these defects, we injected three crRNAs targeting different apln exons (Kroll et al, 2021; Quick et al, 2021) into runx1 mutant zebrafish embryos and control siblings. WISH for myb, a marker for HECs/HSPCs, showed that Apelin signaling loss did not rescue the runx1 mutant phenotype in the CHT (Fig. EV8A–C). Although the possibility of regulatory interactions cannot be excluded, our findings suggest that aplnrb expression is regulated by factor(s) upstream of most pathways previously identified to control HEC specification in the VDA during embryonic development.

## HSPCs derived from converted ECs in apelin mutants persist into adulthood

Upon emergence from the DA, HSPCs enter the circulation and colonize their subsequent niche, the CHT (Murayama et al, 2006). To determine whether the expanded HSPC population in apln mutants can effectively colonize this niche, we analyzed Tg(kdrl:HsHRAS-mCherry); Tg(itga2b:GFP) double transgenic zebrafish larvae by confocal microscopy at 72 hpf. In line with the increased HSPC numbers observed in the DA, MZapln mutants showed an elevated number of HSPCs in the CHT compared to wild-type larvae (Fig. 5A–C). We further validated this increase through WISH for myb at 72 hpf (Fig. 5D,F) and 120 hpf (Fig. 5E,G). The adult hematopoietic niche in the zebrafish is the kidney marrow. To investigate if the expanded HSPC pool is able to

populate the adult HSPC niche, we analyzed hematopoietic cell populations in adult zebrafish kidney marrow by FACS. We could observe increased progenitor cell numbers as well as increased myeloid cell numbers in homozygous apln mutants while erythroid and lymphoid cell numbers in the adult kidney marrow were unaffected by the loss of Apelin signaling (Fig. 5H–J).

## HSPCs derived from converted ECs in apelin mutants have a bias towards the erythroid lineage

Recent studies indicate that HSPC heterogeneity and differentiation potential are inherited from the embryonic endothelium (Xia et al, 2023; Ghersi et al, 2023; Kasper et al, 2020). To assess whether HSPCs in apln mutants display a lineage bias, we conducted WISH to examine marker genes for erythroid (gata1a), lymphoid (rag1) and myeloid (lcp1) lineages. Our results showed a significant expansion of erythroid progenitors in the CHT of homozygous MZapln mutant zebrafish larvae compared to wild-type controls at 4.5 dpf (Fig. 6A–C). In contrast, the numbers of lcp1 expressing myeloid progenitors in the CHT (Fig. 6D–F) and rag1 expressing HSPCs/lymphoblasts in the thymus (Fig. 6G–I) were unaffected by loss of Apelin signaling. Together, these findings suggest that HSPCs in apln mutants not only effectively colonize their niches but also exhibit a bias toward erythroid lineage differentiation.

## Discussion

The embryonic DA exhibits remarkable heterogeneity in EC behaviour and fate. Within the DA, ECs differentiate into three primary fates: (1) angiogenic ECs that sprout from the DA, (2) arterial ECs, that maintain the DA's structural integrity and (3) HECs that give rise to HSPCs. While paracrine signals induce Gata2 and Runx1 expression, EHT and subsequently differentiation into HSPCs (Lam et al, 2010; North et al, 2002; Dobrzycki et al, 2020; Gioacchino et al, 2021; Butko et al, 2015; Kalev-Zylinska et al, 2002), only a subset of ECs undergo this process. This raises the question of what prevents the remaining ECs from transitioning into HECs.

In this study, we identify Apelin signaling is a key regulator of cell fate within the DA. We found that Apelin signaling acts as a molecular switch that balances vascular and hematopoietic identities, with Aplnr expressing ECs biased toward vascular fates, maintaining arterial integrity or angiogenic ECs, while only Aplnr non-expressing ECs are pre-primed to become HECs and give rise to HSPCs. Mechanistically, Apelin signaling inhibits the HEC specification, serving as a gatekeeper between vascular and hematopoietic fates. In its absence, ECs transdifferentiate into HECs, resulting in an expanded hemogenic compartment and increased HSPC numbers. Our data

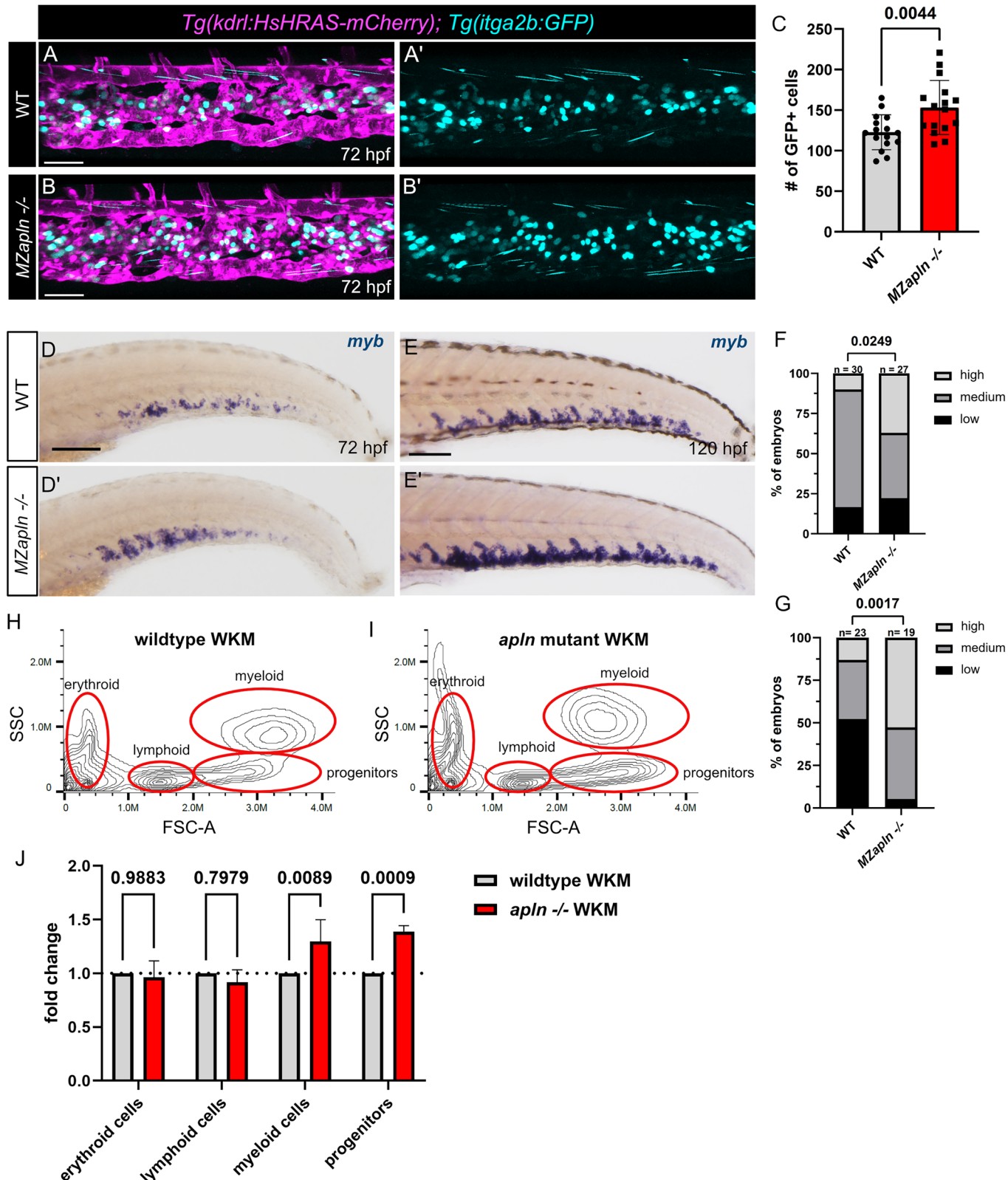

**Figure 5. Expanded hematopoietic stem cell pool can colonize the subsequent stem cell niches.**

(A–B') Confocal projection images of double transgenic *Tg(kdrl:HsHRAS-mCherry)*; *Tg(itga2b:GFP)* zebrafish larvae at 72 hpf. (D, D', E, E') WISH for *myb* on 72 and 120 hpf zebrafish larvae. (A, A') CHT region of a wild-type zebrafish larva. (B, B') CHT region of a *MZapln* −/− mutant zebrafish larva. Elevated HSPC numbers compared to wild-type larvae can be observed. (C) Quantification of *itga2b*:GFP expressing cells in the CHT at 72 hpf, corresponding to (A–B') ($n = 32$, unpaired *t* test, $P = 0.0044$). (D) WISH for *myb* on 72 hpf wild-type larva. (D') WISH for *myb* on 72 hpf *MZapln* −/− mutant larva. Increased *myb* expression compared to wild-type can be observed in *MZapln* −/− mutant zebrafish larvae (E) WISH for *myb* on 120 hpf wild-type larva. (E') WISH for *myb* on 120 hpf *MZapln* −/− mutant larva. Increased *myb* expression compared to wild-type can be observed in *MZapln* −/− mutant zebrafish larvae. (F) Phenotypic distribution plot of larvae in D/D' scored with low, medium and high *myb* expression in the CHT at 72 hpf ($n = 57$, Chi-square test, $P = 0.0249$). (G) Phenotypic distribution plot of larvae in (E-E') scored with low, medium and high *myb* expression in the CHT at 120 hpf ($n = 42$, Chi-square test, $P = 0.0017$). (H, I) Flow cytometry analysis of 4 mpf old dissected whole kidney marrow (WKM) of wild-type and *apln* -/- mutant zebrafish. The hematopoietic lineages are highlighted in red. (J) Quantification of cell populations identified by flow cytometry, corresponding to H/I. ($n = 12$ (WT) and 12 (*apln* -/- mutant). Two-way ANOVA with multiple comparison). CHT caudal hematopoietic tissue, HSPC hematopoietic stem and progenitor cell, WKM whole kidney marrow. Quantifications are displayed as mean ± SD; Scale bars: 50 μm (A–B'); 100 μm (D, D', E, E').

refines the understanding of Apelin signaling during vascular development (Helker et al, 2020). While high Apelin activity promotes angiogenesis, intermediate activity maintains vascular identity, and its absence permits HSPC emergence. These findings position Apelin signaling as a critical modulator of DA cell fate, balancing angiogenesis, vascular maintenance and hematopoiesis. Furthermore, they suggest that pathways controlling vascular maintenance have an impact on lifelong hematopoiesis. Importantly, *aplnr* expression prior to EHT is sufficient to preserve vascular identity at the expense of HEC specification. However, overexpression of *aplnr* in committed HECs does not reverse their hematopoietic trajectory, indicating that Apelin signaling exerts its regulatory influence prior to lineage commitment. Furthermore, we interpret the observed downregulation of the *aplnrb* reporter in hemogenic endothelial cells as occurring shortly after the initiation of *gata2b* expression (Butko et al, 2015). These findings reveal an antagonistic relationship between Apelin signaling and the Gata2-Runx1 transcriptional axis, with Apelin signaling restricting and Gata2-Runx1 signaling promoting HSPC differentiation.

Our findings challenge and refine the limited existing studies on Apelin signaling in hematopoiesis (Yu et al, 2012; Jackson et al, 2021). Previous studies have often assumed a direct role for Apelin signaling in HSC regulation (Yu et al, 2012; Jackson et al, 2021). However, our results demonstrate that this is not the case, as the Aplnr is not expressed in HSPCs. Instead, we reveal that Apelin signaling functions upstream by maintaining vascular identity and restricting HEC specification. This distinction is critical for understanding how Apelin signaling indirectly influences HSPC formation by shaping the endothelial compartment, rather than acting directly on HSPCs. Notably, the downregulation of *Aplnr* mRNA expression during HSPC differentiation has also been observed in mammalian systems (Jackson et al, 2021), supporting the broader relevance of our findings. For example, during in vitro differentiation of mouse embryonic stem cells (ESCs), Aplnr-tdTomato reporter activity was detected in mesodermal cells with hematopoietic potential but declined as differentiation progressed. Although Aplnr-tdTomato expression was detected in CD41+ cells, previous studies relied solely on Apelin ligand stimulation, which showed no effect on HSC numbers (Jackson et al, 2021). These findings align with our work, showing that *aplnrb* is initially expressed in angioblasts during DA formation (Helker et al, 2015) and in ECs in the ventral floor of the DA at 21 hpf, but is subsequently downregulated during HEC differentiation. In the same murine ESC study, stimulation with Apelin ligand during in vitro HSPC differentiation had no effect on the number of CFU-Cs (colony-forming unit cell), despite the presence of Aplnr-tdTomato expression in a subset of CD41 positive cells (Jackson et al, 2021). The discrepancy

between *Aplnr* mRNA and tdTomato expression may be explained by the longer half-life of tdTomato, suggesting that *Aplnr* transcripts may already be absent from these cells. Consistently, two other studies have reported the downregulation of *Aplnr* mRNA expression during HSPC development (Oatley et al, 2020; Bruveris et al, 2020). Furthermore, analysis of published RNAseq data of human embryonic development revealed that the *APLNR* is enriched in arterial endothelial cells, but is among the top downregulated genes in the HE (Calvanese et al, 2022). By employing live imaging of a destabilized fluorophore (Venus-PEST), which has a reduced half-life (Ninov et al, 2012; Li et al, 1998), we visualized *aplnrb* expression at near-endogenous levels in vivo and confirmed that *aplnrb* expression is absent in HECs. The observed downregulation of *Aplnr* during HSC differentiation in mammals supports the broader relevance of our findings, suggesting that the regulatory mechanism described in zebrafish may also apply to other vertebrates, including humans. In summary, these findings indicate that Apelin signaling does not act directly on HSPCs but instead regulates endothelial cell fate decisions upstream of HEC specification.

A fraction of nascent HSPCs proliferates in the VDA (Grainger et al, 2016). However, our work indicates that the elevated HSPC numbers in *apln* mutant embryos are due to enhanced EC-to-HEC conversion, rather than increased proliferation. Furthermore, the resulting HSPCs exhibit lineage bias, favoring erythroid differentiation while myeloid and lymphoid progenitor numbers remain unchanged. Previous studies have shown that HSPC heterogeneity is established before EHT (Xia et al, 2023; Ghersi et al, 2023; Kasper et al, 2020), with microRNAs modulating HSPC lineage potential through cell cycle regulation (Ghersi et al, 2023). Our previous work, along with studies from others, showed that Apelin signaling modulates EC metabolic states (Wilhelm et al, 2016; Helker et al, 2020; De Bock et al, 2013). Given the close relationship between cell cycle regulation and metabolic states, the erythroid bias observed upon loss of Apelin signaling may reflect altered metabolic or cell cycle states in pre-HEC ECs.

In summary, we identify Apelin signaling as a novel upstream regulator of vascular and hematopoietic cell fate within the DA. By precisely controlling the spatio-temporal activity of Apelin signaling, Apelin signaling regulates the balance between vascular maintenance, angiogenesis, and HSPC emergence. Our findings reveal a critical temporal window in which Apelin signaling regulates HEC/HSPC specification and highlights its fundamental role in controlling HSPC numbers and lineage potential. These insights deepen our understanding of the hierarchical acquisition of hematovascular programs and may inform strategies for generating functional HSPCs de novo for therapeutic applications.

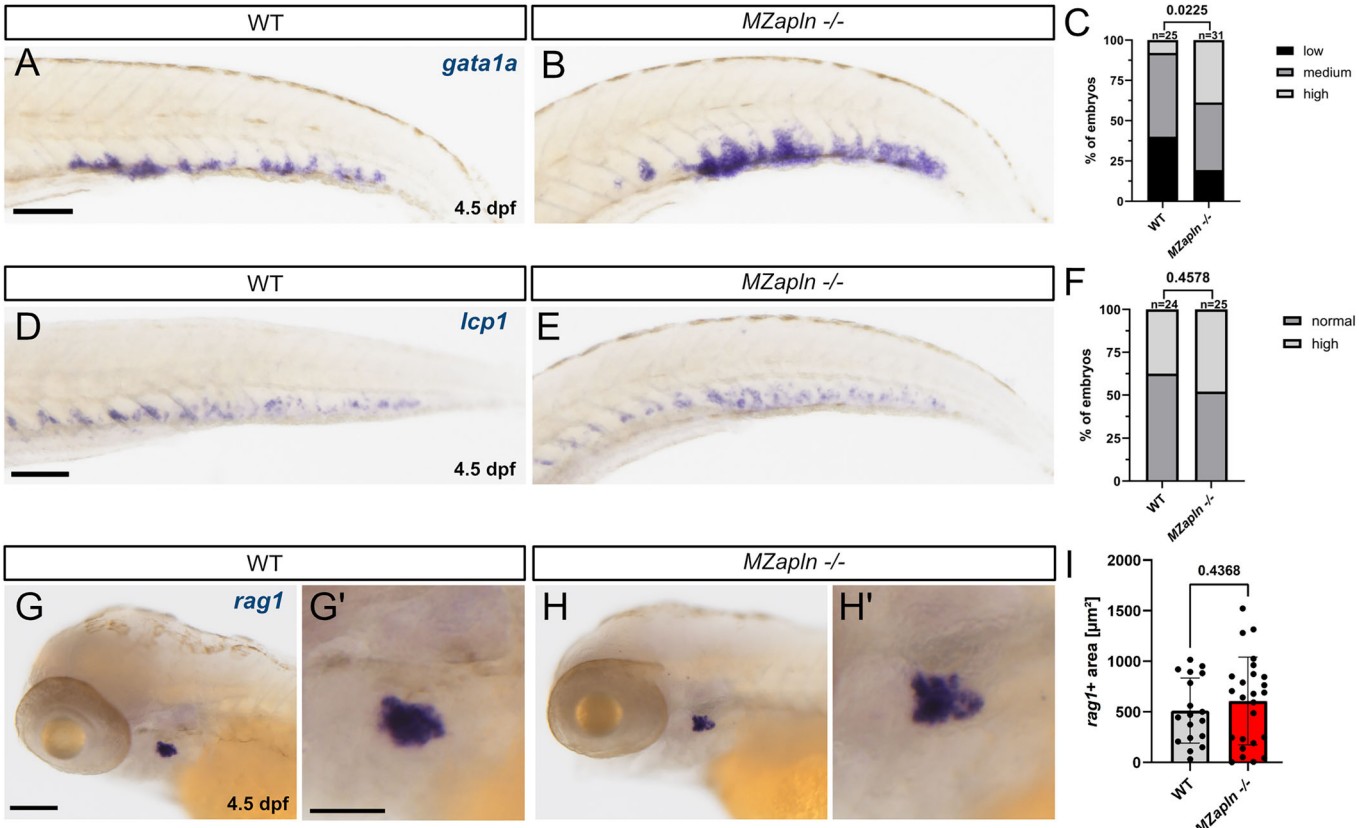

**Figure 6. HSPCs from converted ECs have a bias towards the erythroid lineage.**

(A, B) WISH for *gata1a* at 4.5 dpf in wild-type and *MZapln* −/− mutant zebrafish larvae, showing increased *gata1a* expression in the CHT of *MZapln* −/− mutant larvae. (C) Phenotypic distribution plot of larvae in (A, B) (*n* = 56, Chi-square test, *P* = 0.0225). (D, E) WISH for *lcp1* at 4.5 dpf in wild-type and *MZapln* −/− mutant zebrafish larvae, showing no difference in *lcp1* expression in the CHT. (F) Phenotypic distribution plot of larvae in (D, E) (*n* = 49, Chi-square test, *P* = 0.4578). (G–H′) WISH for *rag1* at 4.5 dpf in wild-type and *MZapln* −/− mutant zebrafish larvae, showing no difference in *rag1* expression in the thymus. (G′–H′) Magnifications corresponding to (G, H). (I) Phenotypic distribution plot of larvae in (G–H′) (*n* = 43, unpaired *t* test, *P* = 0.4368); CHT, caudal hematopoietic tissue; Quantifications are displayed as mean ± SD; Scale bars: 100 μm (A,B,D,E,G,H), 200 μm (G′, H′).

# Methods

### Reagents and tools table

| Reagent/resource | Reference or source | Identifier or catalog number |
|---|---|---|
| **Experimental models** | | |
| *apln*[mu267] (D. rerio) | Helker et al, 2015 | mu267 |
| *aplnrb*[mu281] (D. rerio) | Helker et al, 2020 | mu281 |
| *runx1*[hg1] (D. rerio) | Jin et al, 2009 | hg1 |
| Tg[BAC](aplnrb:Venus-PEST)[mr13] (D. rerio) | Qi et al, 2022 | mr13 |
| Tg(kdrl:NLS-mCherry)[is4] (D. rerio) | Wang et al, 2010 | is4 |
| Tg(kdrl:HsHRAS-mCherry)[s896] (D. rerio) | Chi et al, 2008 | s896 |
| Tg(-6.0itga2b:GFP)[la2] (D. rerio) | Traver et al, 2003 | la2 |
| Tg(fli1:GAL4FF)[ubs4] (D. rerio) | Gutnick et al, 2011 | ubs4 |

| Reagent/resource | Reference or source | Identifier or catalog number |
|---|---|---|
| Tg(gata2b:KALTA4)[sd32] (D. rerio) | Butko et al, 2015 | sd32 |
| Tg(UAS:aplnrb)[mr28] (D. rerio) | This study | mr28 |
| Tg(hsp70l:apln)[mu269] (D. rerio) | Helker et al, 2020 | mu269 |
| Tg (UAS:LIFEACT-EGFP)[mu271] (D. rerio) | Helker et al, 2013 | mu271 |
| Tg(kdrl:GFP)[s843] (D. rerio) | Beis et al, 2005 | s843 |
| TgBAC(apln:EGFP)[bns157] (D. rerio) | Helker et al, 2020 | bns157 |
| Tg(fli1a:GFP)[y1] (D. rerio) | Lawson and Weinstein, 2002 | y1 |
| **Oligonucleotides and other sequence-based reagents** | | |
| qPCR Primer | | |
| *rpl13* se | AATTGTGGTGGTGAGGTG | |
| *rpl13* as | GGTTGGTGTTCATTCTCTTG | |

| Reagent/resource | Reference or source | Identifier or catalog number |
|---|---|---|
| *runx1 se* | CGTCTTCACAAACCCTCCTCAA | |
| *runx1 as* | GCTTTACTGCTTCATCCGGCT | |
| *kdrl se* | GTTCCAGCACCCTTTATCAC | |
| *kdrl as* | GTTTACCATCTTCTCCTACTACAG | |
| *aplnrb se* | CCTCTTGCGCTATGGACTTC | |
| *aplnrb as* | GCCTGCAATCCAGTAGGTCT | |
| crRNAs | | |
| *apln crRNA 1* | CTATGCTCGGTGGAGGCCAT [TGG], Exon 1 | |
| *apln crRNA 2* | GAATGTGAAGATCTTGACGC [TGG], Exon 2 | |
| *apln crRNA 3* | GAAGCATGAGGACTCCTTTG [CGG], Exon 2 | |
| **Chemicals, enzymes and other reagents** | | |
| RO4929097 | MedChemExpress | HY- 11102 |
| DMH-1 | Calbiochem | 203646 |
| LiCl | Sigma-Aldrich | L7026 |
| Purmorphamine | MedChemExpress | HY -15108 |
| **Software** | | |
| GraphPad Prism 10 | GraphPad Software | |
| Imaris 9.7.2 | Oxford Instruments | |
| Fiji (ImageJ). | NIH/Wayne Rasband | |
| NIS Elements | Nikon Instruments | |
| FlowJo 10.7.0 | BD Biosciences | |
| Affinity Designer 2 | Serif (Europe) Ltd | |

## Zebrafish husbandry and strains

All zebrafish housing and husbandry were performed under standard conditions in accordance with institutional (Philipps-Universität Marburg) and national ethical and animal welfare guidelines approved by the ethics committee for animal experiments at the Regierungspräsidium Gießen, Germany, as well as the FELASA guidelines (Aleström et al, 2020) (Akz.: V54-19 c 20 15 f0 2 FB Biologie; V54-19 c 20 15 h 02MR 17/1 Nr. A 1/2020; V54-19 c 20 15 h01MR 17/1 Nr. V8/2022; V54-19 c 20 15 h01MR 17/1 Nr. G 102/2019). Embryos were staged by hours post fertilization (hpf) at 28.5˚C (Kimmel et al, 1995). Fish strains used in this study are listed in the Reagents and tools table.

## Generation of Tg(UAS:aplnrb)^mr28

*UAS:aplnrb* plasmid was generated by cloning the coding sequence of *aplnrb* in *pT2AUAS:lifeact-GFP* (Helker et al, 2013) substituting the *lifeact-GFP* gene. 45 pg plasmid DNA and 300 pg *tol2* mRNA were injected into one-cell stage zebrafish embryos to generate the *Tg(UAS:aplnrb)* zebrafish line. *Tg(UAS:aplnrb)*^mr28 was genotyped by PCR, forward primer: 5'-ATCCTGCAGTGCTGAAAAGC-3'; reverse: 5'- ATTCCCAGTGAGTCCCAGGA-3'.

## FACS sorting and quantitative RT-PCR

The AGM region of 120 double transgenic embryos (*Tg^BAC(aplnrb:Venus-PEST)*^mr13, *Tg(kdrl:NLS-mCherry)*^is4) was manually dissected at 48 hpf and digested with TrypLE Express (gibco) for 30 min (samples were mixed by pipetting every 5 min). Reaction was stopped by adding FBS and cell suspension was pelleted by centrifugation, resuspended in ice-cold HBSS + 5% FBS and poured through 40 μm filters. Centrifugation, resuspension and filtering were repeated once. Cells were FACS sorted for transgenic expression using a BD FACS Aria III (BD Biosciences). Total RNA was isolated using the miRNeasy Micro Kit (Qiagen) and cDNA was synthesized using the iScript™ cDNA Synthesis Kit (Bio Rad). qRT-PCR reaction was performed using an ECO48 real-time qPCR system (PCRmax) with the oligonucleotides listed in the Reagents and tools table.

## Global *apln* overexpression utilizing *Tg(hsp70l:apln)*^mu269

To globally overexpress the Apelin ligand under temporal control, embryos of the *Tg(hsp70l:apln)*^mu269 zebrafish strain and control siblings were subjected to a single heatshock at 37 °C for 1 h at the developmental timepoints indicated in the text.

## Inhibitor treatments

Embryos were treated with the inhibitors/activators listed in the reagents and tools table or DMSO as control in E3 at the indicated timepoints.

## CRISPR injections

Synthetic sgRNAs were obtained from IDT (Coralville, USA). P0 knockouts were done as described (Quick et al, 2021). The target sequences of the used crRNAs are listed in the Reagents and tools table.

## Confocal microscopy

Zebrafish embryos and larvae were raised to the desired developmental stage and mounted in a glass bottom dish in 0.3% agarose containing 20 mg/l Tricaine. Images and time-lapse movies were acquired using an upright Leica ST8 equipped with a temperature-controlled chamber.

## Quantification of HSPCs in the DA and CHT

HSPCs in the VDA were quantified within a segment spanning five somites. HSPCs in the CHT were counted within a segment spanning six somites. Quantification was done by use of Imaris software (Oxford Instruments).

## Whole mount in situ hybridization

Whole mount in situ hybridizations were performed as described (Thisse and Thisse, 2008) using previously published probes (Lundin et al, 2020). Results were documented using a SMZ 18 binocular (Nikon) with a DS-Fi3 camera (Nikon). Embryos and larvae stained for *runx1, myb, gata1a* and *lcp1* were phenotypically scored for low, medium and high expression and quantification was

displayed as a distribution plot. In larvae stained for *rag1* the stained area was measured using Fiji (ImageJ).

## EdU cell proliferation assay

Zebrafish embryos were raised in E3 containing 0.1% PTU until 34 hpf, dechorionated and transferred into 500 μM EdU solution with 10% DMSO in E3. Embryos were incubated in the EdU solution for 1 h on ice and were then allowed to recover for 1 h at 28.5 °C in E3 and fixed in 4% PFA for 1 h at room temperature. EdU labeling was performed according to manufacturer instructions (Click-iT™ Plus EdU Cell Proliferation Kit, Alexa Fluor™ 647 dye, Invitrogen #C10640). Images were recorded with an upright Leica ST8 confocal microscope.

## Analysis of hematopoietic lineages in the whole kidney marrow

For each biological replicate, the kidney marrow of four male adult zebrafish per genotype was dissected at 4 mpf (in total 12 fish per genotype). Hematopoietic lineages were analyzed following the protocol described by Mahony and Monteiro (Mahony and Monteiro, 2024). Cell sorting was performed using the Aurora CS cell sorter (Cytek, Software SpectroFlo). The resulting data were analyzed and visualized using FlowJo (Version 10.7.0). Contour plots with a 5% contour level were used to display the results.

## Statistical analysis

Statistical analysis of the acquired data was performed in Prism 10 (GraphPad). The used statistical tests are stated in the figure legends.

## Large language models (LLMs)

We acknowledge the use of LLMs, specifically ChatGPT, in the preparation of this manuscript. The AI tool was used to assist with language and grammar editing, as well as to refine and improve the clarity of our existing sentences. We emphasize that the LLMs did not generate any "*de-novo*" sentences or results; they were solely used to enhance the presentation of our own original research and writing.

# Data availability

The imaging source data files are deposited in BioImage Archive (Accession number: S-BIAD2382).

The source data of this paper are collected in the following database record: biostudies:S-SCDT-10_1038-S44319-025-00656-6.

# Peer review information

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

## Acknowledgements

We would like to thank Lukas Herdt and Julian Malchow for feedback and comments on the manuscript; Sabine Fischer, Regina Loechel and Hartmann Raifer (FACS Core Facility Marburg) for technical support. The work was supported by the German Research Foundation (Deutsche Forschungsgemeinschaft—DFG) (GRK 2213; Project number 280328342, CSMH) and the National Institute of Diabetes and Digestive and Kidney Diseases (NIDDK, Project number K01DK129409, WWS). Microscopy was performed with the support of the Centre for Advanced Light Microscopy (CALM) Marburg, funded by the German Research Foundation (Deutsche Forschungsgemeinschaft—DFG, Project number 446988475). Open Access funding was provided by the Open Access Publishing Fund of Philipps-University Marburg.

## Author contributions

**Jean Eberlein**: Conceptualization; Investigation; Visualization; Methodology; Writing—original draft; Writing—review and editing. **Nadja Groos**: Investigation; Visualization; Writing—review and editing. **Navina Shrestha Duwal**: Investigation; Writing—review and editing. **Wade W Sugden**: Writing—review and editing. **Trista E North**: Writing—review and editing. **Christian SM Helker**: Conceptualization; Supervision; Funding acquisition; Methodology; Writing—original draft; Project administration; Writing—review and editing.

Source data underlying figure panels in this paper may have individual authorship assigned. Where available, figure panel/source data authorship is listed in the following database record: biostudies:S-SCDT-10_1038-S44319-025-00656-6.

## Funding

## Disclosure and competing interests statement

The authors declare no competing interests.

# Expanded View Figures

**Figure EV1.  *aplnrb* negative cells are restricted to the VDA.**

(A–A″) Confocal projection images of the trunk region of Tg^BAC^(*aplnrb:Venus-PEST*); Tg(*kdrl:NLS-mCherry*) double transgenic zebrafish embryos at 22 hpf. *aplnrb*:Venus-PEST negative ECs can be detected in the ventral DA (cyan arrowheads). (B, B″) Surface rendering of a confocal projection image of a double transgenic Tg^BAC^(*aplnrb:Venus-PEST*); Tg(*kdrl:NLS-mCherry* embryos at 48 hpf highlighting the absence of *aplnrb*:Venus-PEST expression (cyan) from putative HECs (yellow arrowheads) in the ventral DA (nuclei in magenta). (C) Confocal projection image of a double transgenic Tg^BAC^(*aplnrb:Venus-PEST*); Tg(*kdrl:NLS-mCherry* larva at 72 hpf. Expression of *aplnrb*:Venus-PEST is absent from the DA and PCV and is restricted to the intersegmental vessels. (D, D′) Confocal projection images of a Tg^BAC^(*aplnrb:Venus-PEST*); Tg(*kdrl:HsHRAS-mCherry*) double transgenic zebrafish embryo at 48 hpf. *aplnrb*:Venus-PEST negative ECs can be detected in the ventral DA (cyan arrowheads). (E, E′) Sagittal slice of the images in (D, D′). (F, F′) Confocal projection images of the trunk region of a Tg(*kdrl:GFP*); Tg(*kdrl:NLS-mCherry*) double transgenic embryo at 48 hpf. All ECs in the DA are marked with *kdrl*:GFP and *kdrl*:NLS-mCherry expression. (G, G′) Sagittal slice of the images in (F, F′). (H) Quantitative RT-PCR results for *kdrl* on FACS sorted *aplnrb*:Venus-PEST expressing and non-expressing ECs, referring to Fig. 1E–G ($n = 3$, paired $t$ test, $P < 0.0003$). Quantifications are displayed as mean ± SD. (I–J′) Confocal projection images of double transgenic Tg(*kdrl:HsHRAS-mCherry*); Tg^BAC^(*apln:GFP*) zebrafish embryo at 24 hpf (I, I′) and embryo at 48 hpf (J, J′). At both timepoints displayed, no expression of the reporter for *apln* can be detected in the DA and the PCV. The notochord shows *apln* expression that is decreasing over time. DA dorsal aorta, PCV posterior cardinal vein, HECs hemogenic endothelia cells, ECs endothelial cells, NC notochord. Scale bars: 20 µm.

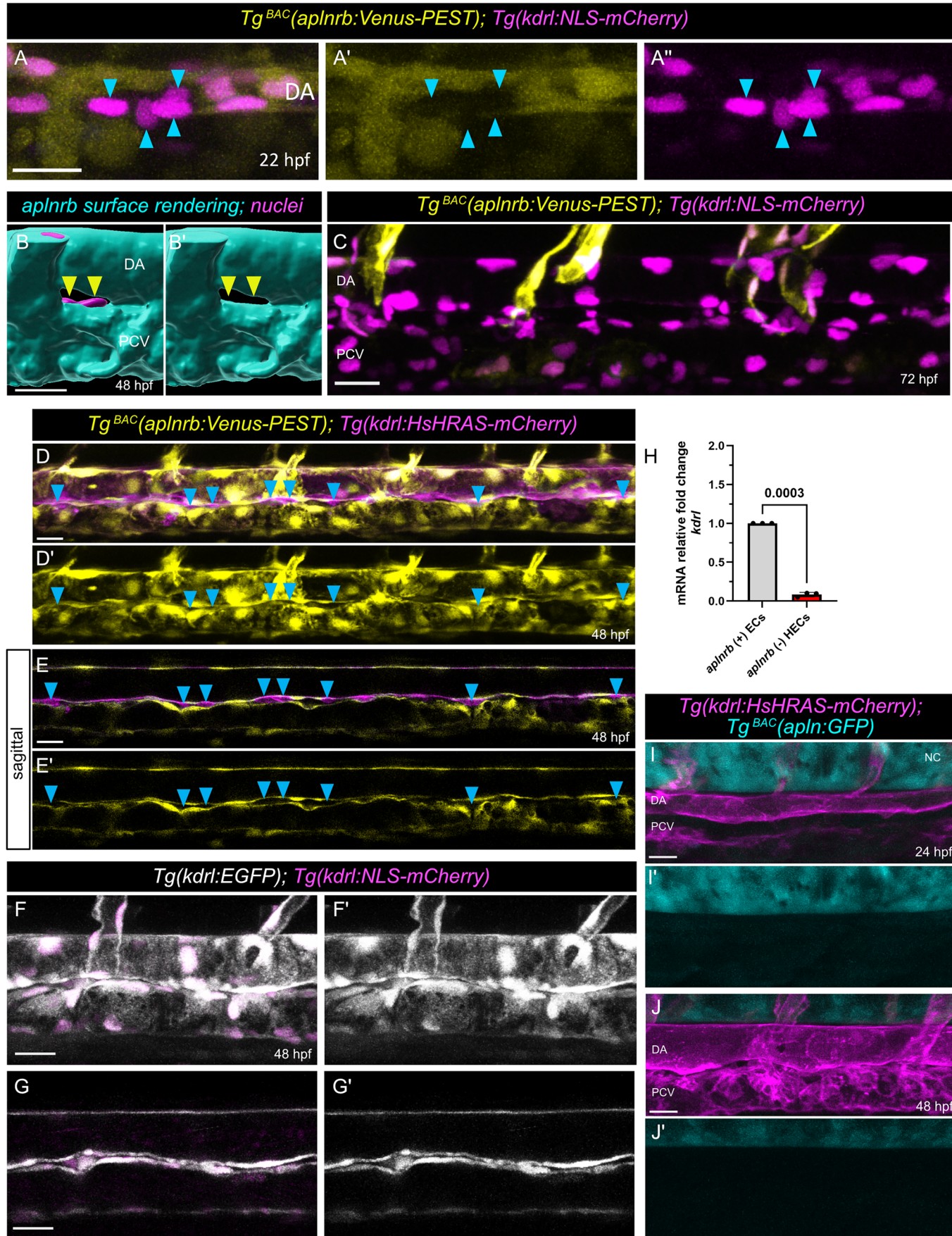

Tg^BAC(aplnrb:Venus-PEST); Tg(kdrl:NLS-mCherry)

aplnrb surface rendering; nuclei

Tg^BAC(aplnrb:Venus-PEST); Tg(kdrl:NLS-mCherry)

Tg^BAC(aplnrb:Venus-PEST); Tg(kdrl:HsHRAS-mCherry)

Tg(kdrl:EGFP); Tg(kdrl:NLS-mCherry)

Tg(kdrl:HsHRAS-mCherry); Tg^BAC(apln:GFP)

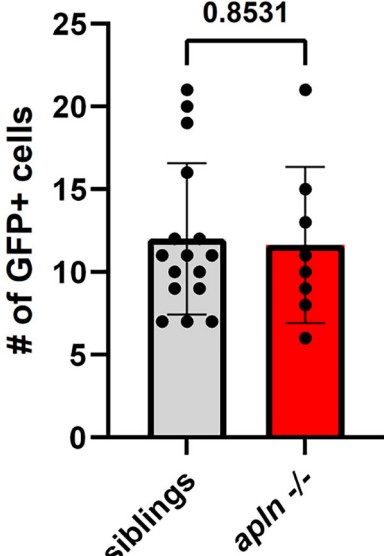

**Figure EV2.   No detectable change in HSPC numbers in zygotic *apln* mutant embryos.**

Quantification of itga2b:GFP expressing cells in the DA at 52 hpf ($n = 32$, unpaired *t* test, $P = 0.0044$); Quantifications are displayed as mean ± SD.

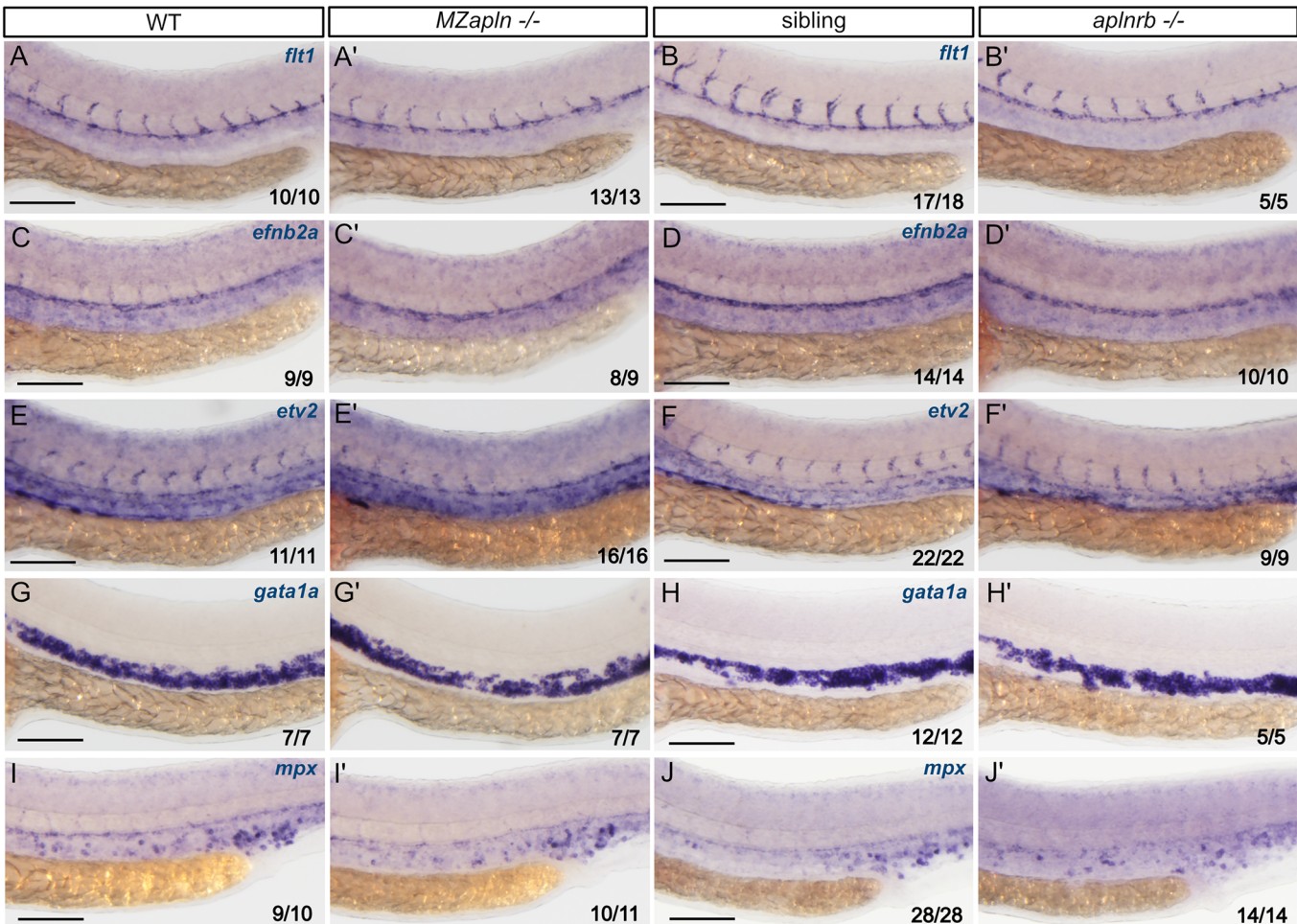

**Figure EV3. Arterial differentiation and primitive hematopoiesis are unaltered upon loss of Apelin signaling.**

(A–J') WISH on wild-type and *MZapln -/-* mutant (A, A', C, C', E, E', G, G', I, I') and siblings and *aplnrb -/-* mutant (B, B', D, D', F, F', H, H', J, J') zebrafish embryos at 24 hpf. (A–B') WISH for *flt1* showing WT-like expression upon loss of Apelin signaling. (C–D') WISH for *efnb2a* showing WT-like expression upon loss of Apelin signaling. (E–F') WISH for *etv2* showing WT-like expression upon loss of Apelin signaling. (G–H') WISH for *gata1a* showing WT-like expression upon loss of Apelin signaling. (I–J') WISH for *mpx* showing WT-like expression upon loss of Apelin signaling. Scale bars: 100 µm.

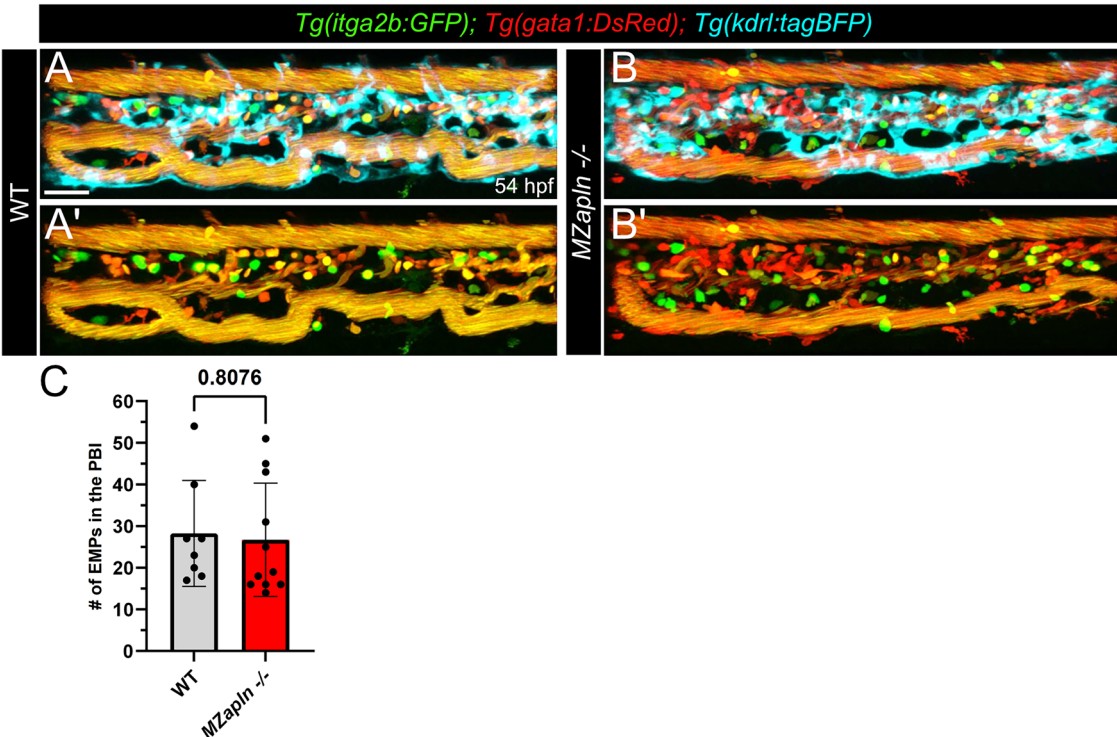

**Figure EV4. EMP development is independent of Apelin signaling.**

(A–B') Confocal projection images of the CHT region of *Tg(itga2b:GFP); Tg(gata1:DsRed); Tg(kdrl:tagBFP)* triple transgenic zebrafish larvae at 54 and 72 hpf. (A–B') Exemplary images of wild-type and *MZapln −/−* mutant zebrafish larvae at 54 hpf. (C) Quantification corresponding to (A–B'). *itga2b*:GFP + /*gata1a*:DsRed+ EMPs were counted in the CHT ($n = 19$, unpaired *t* test, $P = 0.8076$). EMP erythro-myeloid progenitors, CHT caudal hematopoietic tissue. Quantifications are displayed as mean ± SD. Scale bars: 40 μm.

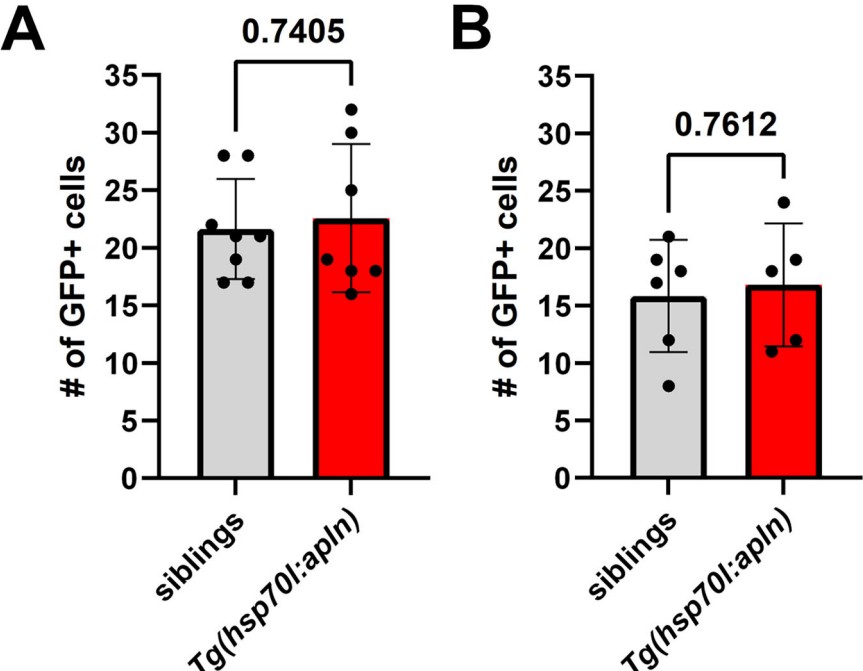

**Figure EV5. Overexpression of *apln* does not influence HSPC specification and maintenance.**

(A, B) Quantification of *itga2b*:GFP expressing HSPCs in the DA at 48 hpf upon heatshock induced global *apln* overexpression. (A) heatshock at 16 hpf, imaging at 48 hpf ($n = 15$, unpaired *t* test, $P = 0.7405$). (B) heatshock at 32 hpf, imaging at 48 hpf ($n = 11$, unpaired *t* test, $P = 0.7612$). HSPC hematopoietic stem and progenitor cell, DA dorsal aorta. Quantifications are displayed as mean ± SD.

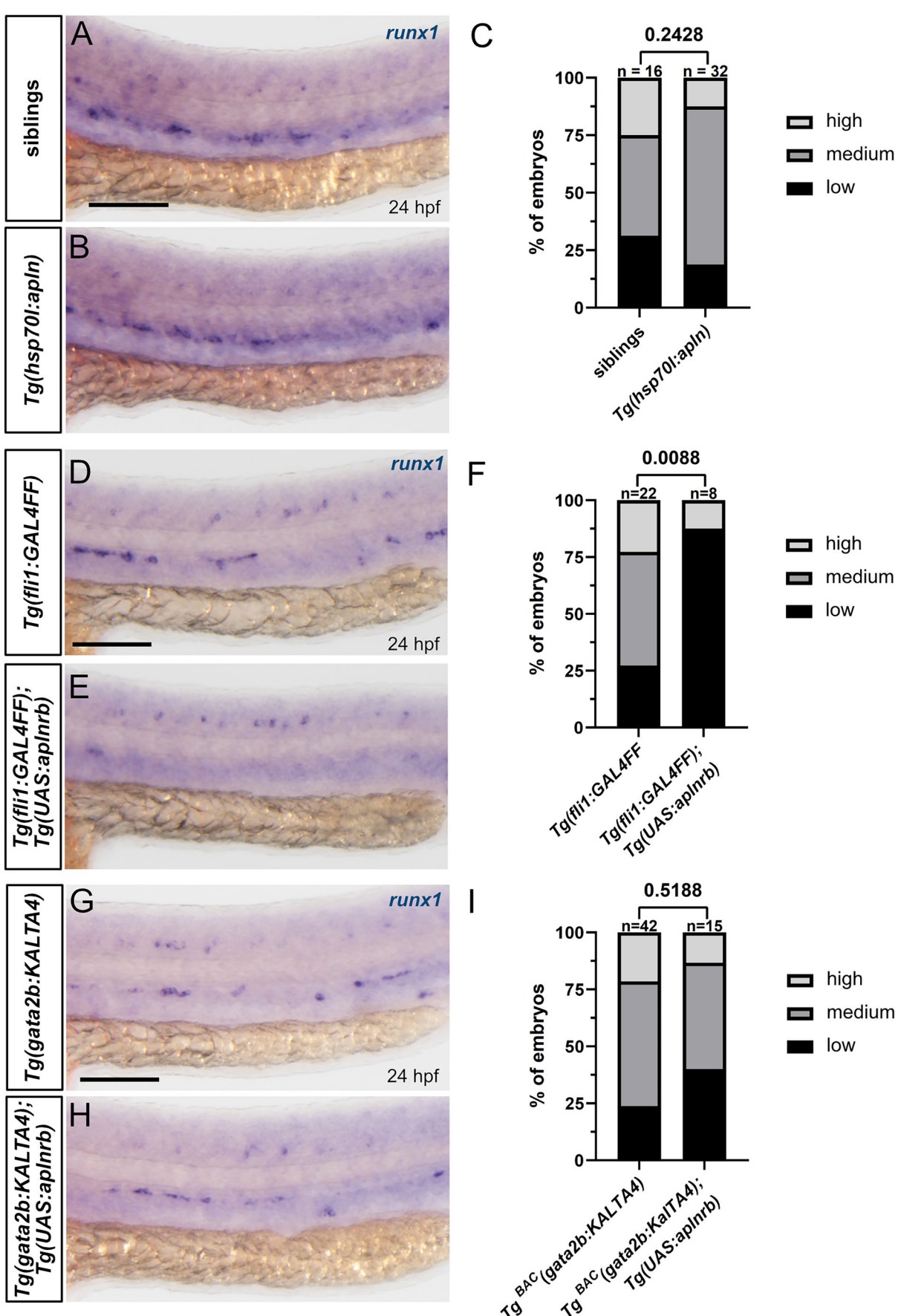

**Figure EV6. Forced expression of *aplnrb* in the vasculature disturbs HE specification.**

(**A, B, D, E, G, H**) WISH for *runx1* at 24 hpf upon *apln/aplnrb* overexpression. Images of the trunk region above the yolk extension. (**A, B**) WISH for *runx1* in siblings or *hsp70l:apln* (heatshock inducible global overexpression of *apln*) zebrafish embryos. The heatshock was induced at 16 hpf. Embryos with global *apln* overexpression display no differences in *runx1* expression compared with control siblings. (**C**) Phenotypic distribution plot of embryos in (**A, B**) scored with low, medium and high *runx1* expression ($n = 48$, Chi-square test, $P = 0.2428$). (**D, E**) WISH for *runx1* in *fli1*:Gal4FF (control) or *fli1*:Galf4FF/*UAS*:aplnrb (pan-vascular *aplnrb* overexpression) zebrafish embryos. Embryos with vascular *aplnrb* overexpression display a drastic reduction in *runx1* expression compared with control siblings. (**F**) Phenotypic distribution plot of embryos in (**D, E**) scored with low, medium and high *runx1* expression ($n = 30$, Chi-square test, $P = 0.0088$). (**G, H**) WISH for *runx1* in *gata2b*:KaltA4 (control) or *gata2b*:KaltA4/*UAS*:aplnrb (hemogenic endothelium specific overexpression of *aplnrb*) zebrafish embryos. Embryos with HE specific *aplnrb* overexpression display no differences in *runx1* expression compared with control siblings. (**I**) Phenotypic distribution plot of embryos in (**G, H**) scored with low, medium and high *runx1* expression ($n = 57$, Chi-square test, $P = 0.5188$). HE hemogenic endothelium. Quantifications are displayed as mean ± SD. Scale bars: 100 μm.

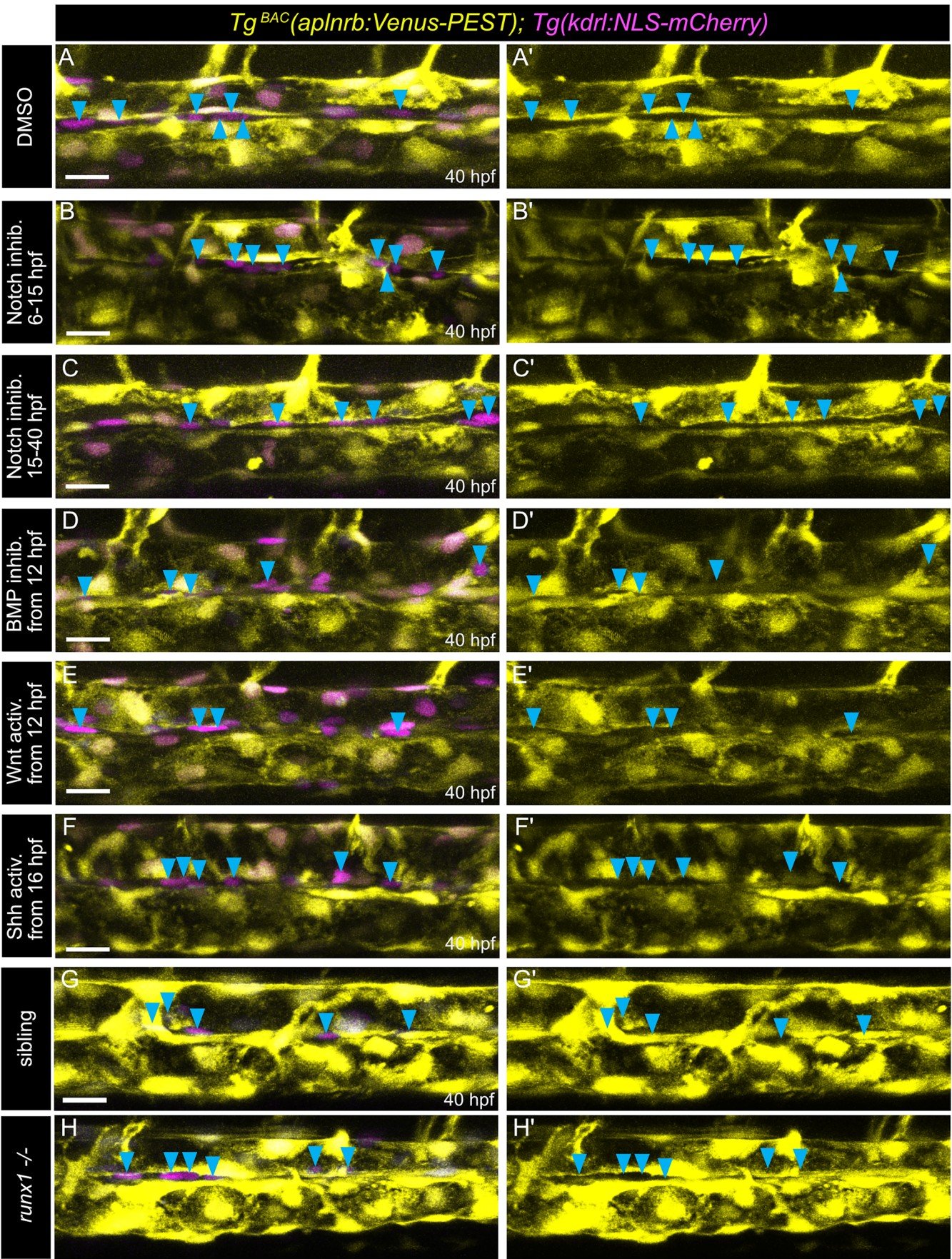

**Figure EV7. *aplnrb* repression in HEC is independent of major regulators of embryonic hematopoiesis.**

(**A–H′**) Confocal projection images of the trunk region of *Tg^BAC^(aplnrb:Venus-PEST); Tg(kdrl:NLS-mCherry)* double transgenic zebrafish embryos. *aplnrb:*Venus-PEST negative HECs can be detected in the ventral DA (cyan arrowheads). (**A, A′**) Trunk region of DMSO treated control embryos corresponding to (**B–F′**), imaging at 40 hpf. (**B, B′**) Trunk region of Notch inhibitor (1 μM RO4929097) treated embryos, treatment from 6–15 hpf, imaging at 40 hpf. WT-like *aplnrb:*Venus-PEST expression in the ventral DA can be observed. (**C, C′**) Trunk region of Notch inhibitor (1 μM RO4929097) treated embryos, treatment from 15–40 hpf, imaging at 40 hpf. WT-like *aplnrb:*Venus-PEST expression in the ventral DA can be observed. (**D, D′**) Trunk region of BMP inhibitor (10 μM DMH1) treated embryos, treatment from 12–40 hpf, imaging at 40 hpf. WT-like *aplnrb:*Venus-PEST expression in the ventral DA can be observed. (**E, E′**) Trunk region of Wnt activator (0.15 M LiCl) treated embryos, treatment from 12–40 hpf, imaging at 40 hpf. WT-like *aplnrb:*Venus-PEST expression in the ventral DA can be observed. (**F–F′**) Trunk region of Shh activator (20 μM Purmorphamine) treated embryos, treatment from 16–40 hpf, imaging at 40 hpf. WT-like *aplnrb:*Venus-PEST expression in the ventral DA can be observed. (**G–H′**) Trunk region of siblings and *runx1* −/− mutant zebrafish embryos at 40 hpf. WT-like *aplnrb:*Venus-PEST expression can be observed in the ventral DA of *runx1* −/− mutant embryos. HE hemogenic endothelium, EC endothelial cell, DA dorsal aorta. Scale bars: 20 μm.

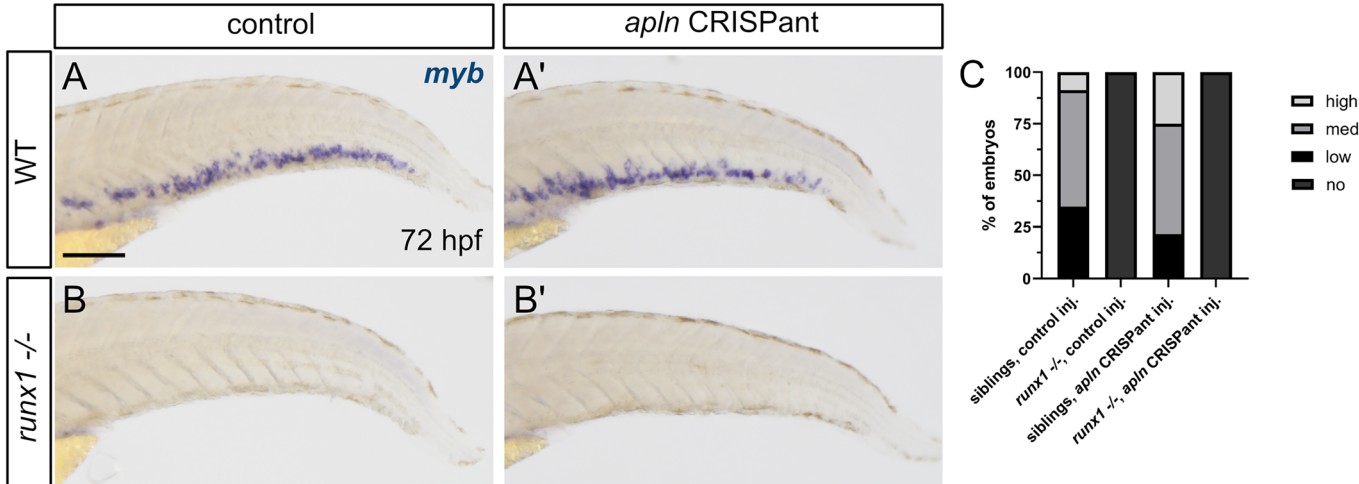

**Figure EV8.   Elevated HSPC numbers upon loss of Apelin signaling are dependent on Runx1 function.**

(A, B) WISH for *myb* at 72 hpf in *runx1 -/-* mutant larvae and siblings. Images of the CHT region. (A, A') WISH for *myb* in control injected and *apln* CRISPR injected wild-type siblings. Loss of Apelin signaling results in increased *myb* expression in the CHT compared with control injected siblings. (B, B') WISH for *myb* in control injected and *apln* CRISPR injected *runx1 -/-* mutant larvae. Loss of Apelin signaling does not rescue *myb* expression in the CHT. (C) Phenotypic distribution plot of embryos in (A, B) scored for no, low, medium and high *myb* expression (*n* = 71). CHT caudal hematopoietic tissue. Scale bars: 200 μm.

                                                