## [Peer Review File · EMBO Reports]

Apelin signaling acts as a molecular switch between endothelial and hematopoietic stem cell fates

Christian Helker, Jean Eberlein, Nadja Groos, Navina Shrestha Duwal, Wade Sugden, and Trista North

Corresponding author(s): Christian Helker (christian.helker@biologie.uni-marburg.de)

Review Timeline:

Submission Date:	8th Mar 25
Editorial Decision:	15th Apr 25
Revision Received:	21st Aug 25
Editorial Decision:	13th Oct 25
Revision Received:	4th Nov 25
Accepted:	17th Nov 25

Transaction Report:

Dear Dr. Helker

Thank you for the submission of your research manuscript to our journal. We have now received the full set of referee reports that is copied below.

As you will see, the referees acknowledge that the findings are interesting and that the conclusions are overall supported by the data presented but they also raise a number of concerns and have suggestions how to further strengthen the data that all seem pertinent and should be addressed.

Given these constructive comments, we would like to invite you to revise your manuscript with the understanding that the referee concerns (as detailed above and in their reports) must be fully addressed and their suggestions taken on board. Please address all referee concerns in a complete point-by-point response. Acceptance of the manuscript will depend on a positive outcome of a second round of review. It is EMBO Reports policy to allow a single round of revision only and acceptance or rejection of the manuscript will therefore depend on the completeness of your responses included in the next, final version of the manuscript.

We realize that it is difficult to revise to a specific deadline. In the interest of protecting the conceptual advance provided by the work, we recommend a revision within 3 months (July 15th). Please discuss the revision progress ahead of this time with the editor if you require more time to complete the revisions.

I am also happy to discuss the revision further via e-mail or a video call, if you wish.

=====
IMPORTANT NOTE:

We perform an initial quality control of all revised manuscripts before re-review. Your manuscript will FAIL this control and the handling will be delayed IN CASE the following APPLIES:

- 1) A data availability section providing access to data deposited in public databases is missing. If you have not deposited any data, please add a sentence to the data availability section that explains that.
- 2) Your manuscript contains statistics and error bars based on $n=2$. Please use scatter blots in these cases. No statistics should be calculated if $n=2$.

=====

- 1) a .docx formatted version of the manuscript text (including legends for main figures, EV figures and tables). Please make sure that the changes are highlighted to be clearly visible.
- 2) individual production quality figure files as .eps, .tif, .jpg (one file per figure). Please download our Figure Preparation Guidelines (figure preparation pdf) from our Author Guidelines pages <https://www.embopress.org/page/journal/14693178/authorguide> for more info on how to prepare your figures.
- 3) a .docx formatted letter INCLUDING the reviewers' reports and your detailed point-by-point responses to their comments. As part of the EMBO Press transparent editorial process, the point-by-point response is part of the Review Process File (RPF), which will be published alongside your paper.
- 4) a complete author checklist, which you can download from our author guidelines

(<<https://www.embopress.org/page/journal/14693178/authorguide>>). Please insert information in the checklist that is also reflected in the manuscript. The completed author checklist will also be part of the RPF.

5) Please note that all corresponding authors are required to supply an ORCID ID for their name upon submission of a revised manuscript (<<https://orcid.org/>>). Please find instructions on how to link your ORCID ID to your account in our manuscript tracking system in our Author guidelines (<<https://www.embopress.org/page/journal/14693178/authorguide#authorshipguidelines>>)

6) We replaced Supplementary Information with Expanded View (EV) Figures and Tables that are collapsible/expandable online. A maximum of 5 EV Figures can be typeset. EV Figures should be cited as "Figure EV1, Figure EV2" etc... in the text and their respective legends should be included in the main text after the legends of regular figures.

7) Please include a dedicated "Data Availability" section at the end of the Methods (suggested wording: "The [structural coordinates | microarray | mass spectrometry] data from this publication have been deposited to the [name of the database] database [URL] and assigned the identifier [accession | permalink | hashtag]."). Should this not apply, this should still be stated as "This study includes no data deposited in external repositories."

Additional information on source data and instruction on how to label the files are available <<https://www.embopress.org/page/journal/14693178/authorguide#sourcedata>>

10) Figure legends and data quantification:
The following points must be specified in each figure legend:

- the name of the statistical test used to generate error bars and P values,
 - the EXACT p-values,
 - the number (n) of independent experiments (please specify technical or biological replicates) underlying each data point,
 - the nature of the bars and error bars (s.d., s.e.m.)
- If the data are obtained from n {less than or equal to} 5, show the individual data points in addition to the SD or SEM.
- If the data are obtained from n {less than or equal to} 2, use scatter blots showing the individual data points.

See also the guidelines for figure legend preparation:
<https://www.embopress.org/page/journal/14693178/authorguide#figureformat>

11) Our journal encourages inclusion of *data citations in the reference list* to directly cite datasets that were re-used and obtained from public databases. Data citations in the article text are distinct from normal bibliographical citations and should directly link to the database records from which the data can be accessed. In the main text, data citations are formatted as follows: "Data ref: Smith et al, 2001" or "Data ref: NCBI Sequence Read Archive PRJNA342805, 2017". In the Reference list, data citations must be labeled with "[DATASET]". A data reference must provide the database name, accession number/identifiers and a resolvable link to the landing page from which the data can be accessed at the end of the reference. Further instructions are available at <<https://www.embopress.org/page/journal/14693178/authorguide#referencesformat>>.

12) All Materials and Methods need to be described in the main text using our 'Structured Methods' format. According to this

format, the Methods section includes a Reagents and Tools Table (listing key reagents, experimental models, software and relevant equipment and including their sources and relevant identifiers) followed by a Methods and Protocols section describing the methods, ideally using a step-by-step protocol format. The aim is to facilitate adoption of the methodologies across labs. Please download and fill our Reagents and Tools Table template (.docx), which you can find in our author guidelines: <https://www.embopress.org/page/journal/14693178/authorguide#structuredmethods>.

13) As part of the EMBO publication's Transparent Editorial Process, EMBO Reports publishes online a Review Process File to accompany accepted manuscripts. This File will be published in conjunction with your paper and will include the referee reports, your point-by-point response and all pertinent correspondence relating to the manuscript.

Yours sincerely,

=====

Referee #1:

Comments Eberlein et al

In this manuscript the authors identify a new role for apelin signalling in regulating endothelial cell fate between angiogenic and haemogenic cell fate. This is mediated by *aplnrb*, present in ECs but absent from HEC. By using confocal imaging and clever genetics, they go on to show that absence of the *aplnrb* leads to increased HSPC numbers, not by proliferation but likely by fate conversion from (arterial) ECs. For the most part, the conclusions are well supported by the data. Overall, this is an elegant piece of work that clarifies further what the role of apelin signalling is in helping to direct HEC cell fate. However, I suggest that a few more controls, a more appropriate timing for some experiments and some clarification regarding statistics and the experiments in adults are required to strengthen the conclusions from these experiments prior to publication.

Major comments

1. The qPCR experiments looking at *runx1* and *aplnrb* are performed at 48hpf. However, we'd expect arterial EC to HEC fate decisions to happen much earlier, perhaps more obviously from 26-30hpf and before any EHT occurs. This is the correct stage to perform this qPCR and you might find a bigger difference in expression levels then.
2. The analysis of the EMP numbers performed at 3dpf has a similar problem; the peak of EMP cell numbers is at 48hpf, not 3dpf (Bertrand et al paper). At 3dpf, the *gata1-DsRed* transgenic might be reading HSPC-derived erythroid progenitors in addition to any EMPs. By 48hpf this should not be an issue.
3. The analysis of in situ experiments in the manuscript (e.g. Fig 2F,I; Fig 5F,G, etc) could be improved by measuring pixel intensities rather than binning signal strength into categories.
4. Testing known pathways for modulation of *aplnrb* expression is a good experiment but is missing a control experiment (e.g. by in situ hybridization to *runx1* and/or *cmyb* to show that the treatments induce the outcomes expected from the publications cited). Related to that, it's unclear to me what percentage of HECs are *aplnrb*-negative. Is it all of them? Some?
5. Fig 5 - from the text, the authors claim an increase in adult HSPCs and a slight increase in erythroid cell counts (Fig 5H-J). While we are shown percentages of cell populations defined by FSC/SSC in panel J, no statistics is shown to backup the

assertion in the text - Showing these stats should strengthen their argument. In addition, it's not clear from the text how HSPCs were measured in these adults - was it measured indirectly from the progenitor gate? HSPCs would be located in the lymphoid and progenitor gates, not just the progenitors, so the claimed changes could reflect changes in other, more restricted progenitor types (e.g. CMPs). A better measure of HSPC numbers would be CD41-GFP+ cells from whole marrow.

Minor comments

1. Introduction, line 22 - as you will know, *itga2b* (at least by transgene expression), does not initiate before EHT starts (so from 32-324hpf onwards), a few hours after *runx1* and *cmyb* are expressed in the DA (~23hpf)
2. Fig. 1F,1G - add gene name to the panels to make the figure easier to read.
3. Check the order of the supplementary figures against the text. There are some instances where the text calls for supplementary figures out of place. For example, EV2 is referred to (line13, p7) but doesn't relate to the text there, while EV3 does. Perhaps best to swap those two for clarity. There are similar issues with figures EV5 and EV6, for example, where a change in the figures to match the text would improve readability.
4. Did you independently verify the gRNA activity/efficiency in targeting *apln*?

Referee #2:

This is an excellent manuscript that provides novel mechanistic insight into a highly interesting biological question. The authors show, in a series of very high quality, conclusive experiments, that the level of apelin receptor expression determines whether endothelial cells will transdifferentiate into hematopoietic stem and progenitor cells (HSPCs) during zebrafish development. The zebrafish embryo is a marvelous model to explore the specification of HSPCs, and the authors make full use of the superior imaging capabilities in the model. By combining imaging with genetic interventions, their work clarifies and refines previously published models about the role of Apelin signaling in HSPC development, and is thus of high value. While the data are generally of excellent quality and very clearly presented, I would recommend a few more controls to strengthen the conclusions even further as detailed below. If the authors can provide these additional controls, I can fully recommend publication in EMBO Rep.

1. Fig.3: while the tissue-specific overexpression of the apelin receptor by the Gal4-UAS system is very nice, there is an additional control that the authors could do to make the conclusions drawn for the data even stronger. That is, it is theoretically possible that expression does not work in all tested cells, e.g. because the UAS responder line is not activatable (e.g. due to positional effects) in the HECs, but only the endothelial cells. This lack of expression in HECs would invalidate the author's conclusion that receptor overexpression specifically in the endothelial cells suppresses HSPC formation. Thus, the authors could show by in situ hybridization that both combinations of lines result in receptor overexpression in the targeted cells.
2. Fig 3: GFP in most cells labelled with arrowheads is very dim and hardly visible, thus it is unclear how the cells labelled with arrowheads were called positive. Authors should improve brightness / contrast.
3. Fig. 3: please clarify whether siblings to the *hsp70l:apln* transgenics were heat-shocked as well, which is the appropriate control for heat-shock experiments.
4. Fig. EV7: did those drug treatments (Notch inhibitor, Shh activator, Wnt activator) produce other phenotypes? In the absence of such positive controls, the authors should not conclude that they do not affect *apln* expression.
5. Fig. 5H-J: can the authors give measures of reproducibility / variability for these data? Since the differences between wild-type and mutants is small and the data is pooled from just 4 fish each, in the absence of such data it is difficult to conclude that the observed differences are reproducible / statistically significant.

Referee #3:

Eberlin et al. first identify that, over the course of 20 to 24 hpf, certain endothelial cells (marked by HRAS and KDR) in the ventral dorsal aorta lose apelin receptor expression. Given the location and developmental timing of these cells, the authors speculated that the cells may be contributing to the endothelial to hematopoietic transition (EHT). Upon sorting and qPCR of APLNRB-expressing and non-expressing cells, APLNRB non-expressing cells had higher levels of RUNX1 and lower levels of KDRL. The authors crossed the *aplnrb* reporter zebrafish with *itga2b* expressing cells, a marker of nascent hematopoietic stem and progenitor cells (HSPCs), and observed that apelin receptor non-expressing cells, also express the hematopoietic marker ITGA2B (CD41). Using time lapse imaging, the authors say that they can visualize the APLNR-negative cells budding into the dorsal aorta.

The authors then perform a series of experiments that eliminate expression of either apelin or the apelin receptor and examine

the number of itga2b+ cells and/or runx1 expression. The data suggests that elimination of apelin signalling increases the number of hematopoietic cells and runx1 expression. In contrast, overexpression of apln does not decrease hematopoietic output, but overexpression of the aplnr^b does, indicating that it is the lack of aplnr^b expression on these cells that is supportive of the hematopoietic state. Finally, the knockout of apelin using CRISPRs increases the number of HPSCs in the VDA, with some cells co-expressing gata2b and aplnr^b, indicating that, if apelin signalling were present, these cells would likely remain in an endothelial state.

Mechanistically, aplnr^b knockout cells do not have higher proliferation, indicating that this increase in hematopoietic progenitors is not simply due to a differential proliferation phenotype. Further, they use small molecule inhibitors to show that inhibition of Notch, BMP4, Wnt or Sonic hedgehog activity does not impact apelin receptor expression. Aplnr^b overexpression from gata2b or runx1 promoters, show no difference in HSPC emergence, while knockout of runx1 has no impact on apelin signalling. This evidence indicates that apelin signalling, or lack thereof, is important prior to the EHT, in the initial endothelial commitment stage.

Overall, the authors show that modulation of apelin signalling early in development can alter HSPC emergence. Understanding the timing and the role of apelin is informed by the studies performed, but questions remain and interpretation is challenging.

Major Comments:

The authors use of a short-lived fluorescent reporter Venus-PEST enables more specific quantification of the kinetics of this process, a strength of this paper. However, the timeline of this process is quite confusing. While they start to see downregulation of aplnr^b at 22hpf, they also mention in the introduction that gata2b, the earliest marker of the EHT, can be detected at 20hpf. Thus, does this process happen after the onset of the EHT, following gata2b expression? Timelines representing the experimental frameworks and to summarize the findings at the end would help to increase comprehension in this area.

Many of the images provided are hard to interpret and unconvincing as evidence for general conclusions. Please provide quantification to support all main conclusions discussed in the text. For instance, the conclusion that aplnr^b downregulation is needed prior to the EHT is unfounded as there is no quantification of how many of the budding cells arise from HEC that have downregulated aplnr^b. While the images suggest that there is budding occurring from aplnr^b- cells, it can be hard to interpret the images provided. Quantification such as the percentage of cells budding from aplnr^b- vs aplnr^b+ cells would supplement these conclusions and give context to the requisite nature of this aplnr^b downregulation.

Minor Comments:

The use of zebrafish genetic terminology, while required, is hard to follow. Improvement of readability would help readers understand the significance of this work.

There is very little information about apelin signalling mechanisms. While the authors show apelin receptor b and apelin expression, and that modulation of those components can alter the number of budding cells, they do not ever show active signalling within the cells still possessing the apelin receptor. Downstream target investigation from the apelin receptor would help to understand how the signalling alters the current thinking about the EHT requirements. Given that mechanism is not needed for the scope of this journal, adding to the introduction and discussion about what is known about apelin signalling would help to situate how the signalling of this process integrates with current knowledge of the EHT.

Please ensure the statistical tests used are the appropriate choices. For example, in the FACS of aplnr^b expressing cells, the authors used an unpaired t-test to compare the two conditions. In this reviewer's experience, these experiments are usually paired, with each sort representing a biological replicate. If this is the case, a paired t-test would be a more appropriate comparison.

The erythroid cells from apln mutants have much different side scatter profiles than those in wild type. Does this indicate a different phenotype of those erythroid cell type?

In Galvanese et al. (2022) "Mapping Human Hematopoietic Cells from Hemogenic Endothelium to Birth", there is full sequencing of the human EHT process. Confirming the expression of aplnr^b and apln throughout this process would help to solidify its relevance within the human system.

Martina Rembold, Ph.D.
Senior Editor
On behalf of EMBO reports

August 21, 2025

Dear Martina,

Please find attached our revised manuscript entitled “**Apelin signaling acts as a molecular switch between endothelial and hematopoietic stem cell fates**” for consideration as a Research Article in *EMBO Reports*. We thank you for the initial positive feedback and for giving us the opportunity to resubmit.

We deeply appreciate your constructive comments and those of the reviewers, and have now made significant changes to the manuscript. We have included the results of several new experiments that considerably strengthen the conclusions reached in our manuscript. In particular, we have added/exchanged panels in Figure 1, Figure 5 and Figure EV 1. To improve readability, we changed the order of supplementary figures EV2 and EV3 as well as EV5 and EV6.

Please find below our detailed point-by-point response to the reviewers' comments. All major changes in the main text have been highlighted.

Thank you very much for your consideration of our revised manuscript.

Best regards,
Christian Helker

Reviewer #1

General comment

*In this manuscript the authors identify a new role for apelin signalling in regulating endothelial cell fate between angiogenic and haemogenic cell fate. This is mediated by *aplnrb*, present in ECs but absent from HEC. By using confocal imaging and clever genetics, they go on to show that absence of the *aplnrb* leads to increased HSPC numbers, not by proliferation but likely by fate conversion from (arterial) ECs. For the most part, the conclusions are well supported by the data. Overall, this is an elegant piece of work that clarifies further what the role of apelin signalling is in helping to direct HEC cell fate. However, I suggest that a few more controls, a more appropriate timing for some experiments and some clarification regarding statistics and the experiments in adults are required to strengthen the conclusions from these experiments prior to publication.*

Major comments:

*1. The qPCR experiments looking at *runx1* and *aplnrb* are performed at 48hpf. However, we'd expect arterial EC to HEC fate decisions to happen much earlier, perhaps more obviously from 26-30hpf and before any EHT occurs. This is the correct stage to perform this qPCR and you might find a bigger difference in expression levels then.*

We thank the Reviewer for the valid suggestions. We agree that the arterial EC-to-HEC fate decision occurs earlier, typically between 26–30 hpf. However, the aim of our qPCR experiment was not to capture the earliest transcriptional changes during HEC specification, but rather to determine whether *aplnrb*-negative hemogenic endothelial cells give rise to HSPCs. Since 48 hpf corresponds to the peak of endothelial-to-hematopoietic transition (EHT) and HSPC emergence (Bonkhofer et al. 2019), we selected this time point to best capture changes in *runx1* and *aplnrb* expression relevant to functional HSPC output.

*2. The analysis of the EMP numbers performed at 3dpf has a similar problem; the peak of EMP cell numbers is at 48hpf, not 3dpf (Bertrand et al paper). At 3dpf, the *gata1-DsRed* transgenic might be reading HSPC-derived erythroid progenitors in addition to any EMPs. By 48hpf this should not be an issue.*

We acknowledge that by 3 dpf, the *gata1:DsRed* transgene may label both EMP- and HSPC-derived erythroid progenitors, potentially confounding the analysis. However, our study also includes data at 54 hpf (see Fig. EV4 A, B, E). Therefore, our analysis accounts for the developmental window during which EMPs can be reliably assessed without interference from later hematopoietic lineages.

*3. The analysis of *in situ* experiments in the manuscript (e.g. Fig 2F,I; Fig 5F,G, etc) could be improved by measuring pixel intensities rather than binning signal strength into categories.*

We appreciate the Reviewer's thoughtful suggestion to quantify *in situ* hybridization (ISH) signals using pixel intensities, and we recognize the value and precision of this approach. However, in our experimental setup, we observe considerable biological variation in the number and spatial distribution of HSPCs, which would confound intensity-based measurements and reduce interpretability. Given that *in situ* hybridization is inherently semi-quantitative, we chose categorical scoring as a robust and reproducible method to capture biologically relevant differences while minimizing technical variability from staining and imaging. This approach aligns with methods used in comparable studies (Lundin et al. 2020; Soto et al. 2021) and is well-suited to our study's goals. We nonetheless acknowledge the strengths of intensity-based

quantification and will consider it in future analyses where cell number variability is less of a confounding factor.

4. Testing known pathways for modulation of *aplnrb* expression is a good experiment but is missing a control experiment (e.g. by *in situ* hybridization to *runx1* and/or *cmyb* to show that the treatments induce the outcomes expected from the publications cited).

We thank the Reviewer for this valuable suggestion. All chemical treatments were validated by established phenotypic readouts: RO4929097 reduced expression of the Notch reporter *Tg(EPV.Tp1-Mmu.Hbb:Venus-Mmu.Odc1)s940* (Ninov, Borius, and Stainier 2012) and led to a strong reduction in *runx1* expression compared to controls (Fig. R1A–C); DMH1 disrupted caudal vein plexus formation (Nakajima et al. 2023) and similarly resulted in a strong reduction in *runx1* expression (Fig. R1A–C); LiCl treatment caused loss of eye structures (Van De Water et al. 2001) and led to a modest increase in *runx1* expression (Fig. R1D); Purmorphamine caused pooling of primitive blood in the posterior blood island (Wilkinson et al. 2009) but did not affect *runx1* expression (Fig. R1E).

Fig R1 *runx1* ISH after inhibitor/activator treatments

(A-E) WISH for *runx1* at 24 hpf upon chemical compound treatment. **(A)** Trunk region of DMSO treated control embryos. **(B)** Trunk region of Notch inhibitor (1 μ M RO4929097) treated embryos, treatment from 12 – 24 hpf. **(C)** Trunk region of BMP inhibitor (10 μ M DMH1) treated embryos, treatment from 12 - 40 hpf. **(D)** Trunk region of Wnt activator (0.15M LiCl) treated embryos, treatment from 12 - 24 hpf. **(E)** Trunk region of Shh activator (20 μ M Purmorphamine) treated embryos, treatment from 12 - 24 hpf.

*Related to that, it's unclear to me what percentage of HECs are *aplnrb*-negative. Is it all of them? Some?*

We also thank the reviewer for this valid question. To address it, we analyzed nascent HSPCs and found that nearly all HSPCs in the ventral wall of the dorsal aorta (VDA) originate from *aplnrb*-non-expressing HECs (mean = 99.18%, $n = 23$ embryos). These findings have now been included in Figure 1J.

5. Fig 5 - from the text, the authors claim an increase in adult HSPCs and a slight increase in erythroid cell counts (Fig 5H-J). While we are shown percentages of cell populations defined by FSC/SSC in panel J, no statistics is shown to backup the assertion in the text - Showing these stats should strengthen their argument. In addition, it's not clear from the text how HSPCs were measured in these adults - was it measured indirectly from the progenitor gate? HSPCs would be located in the lymphoid and progenitor gates, not just the progenitors, so the claimed changes could reflect changes in other, more restricted progenitor types (e.g. CMPs). A better measure of HSPC numbers would be CD41-GFP+ cells from whole marrow.

We thank the Reviewer for this constructive feedback. We repeated the whole kidney marrow (WKM) FACS analysis in three independent experiments, using four pooled WKMs per genotype per replicate. Zebrafish dissection, sample preparation, and gating strategy to define hematopoietic cell populations were performed as in the recent standard in the field according to Mahony and Monteiro (2024). These analyses confirmed a significant increase in both myeloid and progenitor cell populations (Fig. 5H–J). Originally, HSPC percentages were measured indirectly from the progenitor gate. To avoid overinterpretation, we have revised the text and now refer to this population more accurately as “progenitor cells” rather than “HSPCs.” Additionally, we have updated Figure 5J to present fold change over WT instead of raw percentages, thereby improving the clarity and comparability of the data.

Minor comments:

1. Introduction, line 22 - as you will know, *itga2b* (at least by transgene expression), does not initiate before EHT starts (so from 32-324hpf onwards), a few hours after *runx1* and *cmyb* are expressed in the DA (~23hpf)

We thank the reviewer for this legitimate point. We altered the phrasing in the text p.2, line 22.

2. Fig. 1F,1G - add gene name to the panels to make the figure easier to read.

We thank the reviewer for this valid suggestion. To improve readability of the figure we added the gene names above the figure panels Fig. 1, F and G.

3. Check the order of the supplementary figures against the text. There are some instances where the text calls for supplementary figures out of place. For example, EV2 is referred to (line13, p7) but doesn't relate to the text there, while EV3 does. Perhaps best to swap those two for clarity. There are similar issues with figures EV5 and EV6, for example, where a change in the figures to match the text would improve readability.

We thank the reviewer for this legitimate note and changed the order of supplementary figures EV2 and EV3 as well as EV5 and EV6. Minor adjustments to the text have been made to accommodate the swapped figures.

4. Did you independently verify the gRNA activity/efficiency in targeting *apln*?

We thank the Reviewer for this valid question. The *apln* CRISPRants reliably phenocopy the intersegmental vessel (ISV) defects observed in homozygous *apln* mutant embryos from the zebrafish line *apln^{mu267}*. This well-characterized line has been used in multiple peer-reviewed studies investigating Apelin signaling (Helker et al. 2020; Herdt et al. 2025; Kwon et al. 2016; Malchow et al. 2024; Qi et al. 2022) and it also serves as the

genetic background for the majority of experiments presented in this manuscript. These consistent phenotypes support the specificity and reliability of the CRISPR-mediated loss-of-function approach used in our study.

Reviewer #2

General comment

This is an excellent manuscript that provides novel mechanistic insight into a highly interesting biological question. The authors show, in a series of very high quality, conclusive experiments, that the level of apelin receptor expression determines whether endothelial cells will transdifferentiate into hematopoietic stem and progenitor cells (HSPCs) during zebrafish development. The zebrafish embryo is a marvelous model to explore the specification of HSPCs, and the authors make full use of the superior imaging capabilities in the model. By combining imaging with genetic interventions, their work clarifies and refines previously published models about the role of Apelin signaling in HSPC development, and is thus of high value. While the data are generally of excellent quality and very clearly presented, I would recommend a few more controls to strengthen the conclusions even further as detailed below. If the authors can provide these additional controls, I can fully recommend publication in EMBO Rep.

Major comments:

1. Fig.3: while the tissue-specific overexpression of the apelin receptor by the Gal4-UAS system is very nice, there is an additional control that the authors could do to make the conclusions drawn for the data even stronger. That is, it is theoretically possible that expression does not work in all tested cells, e.g. because the UAS responder line is not activatable (e.g. due to positional effects) in the HECs, but only the endothelial cells. This lack of expression in HECs would invalidate the author's conclusion that receptor overexpression specifically in the endothelial cells suppresses HSPC formation. Thus, the authors could show by *in situ* hybridization that both combinations of lines result in receptor overexpression in the targeted cells.

We thank the Reviewer for this constructive suggestion. To confirm the functionality of the Gal4-driver lines and the Tg(UAS:aplnrb) effector line, we performed qRT-PCR on dissected trunks from embryos with the following genotypes:

- (1) *fli:Gal4FF; UAS:lifeact-GFP* (control);**
- (2) *fli:Gal4FF; UAS:lifeact-GFP; UAS:aplnrb* (pan-endothelial overexpression);**
- (3) *gata2b:KalTA4; UAS:lifeact-GFP* (control); and**
- (4) *gata2b:KalTA4; UAS:lifeact-GFP; UAS:aplnrb* (HE/HSPC-specific overexpression).**

qRT-PCR analysis revealed a 6.2-fold increase in *aplnrb* expression in the pan-endothelial overexpression group and a 2-fold increase in the HE/HSPC-specific overexpression group (Fig. R2). Given the relatively small number of HECs/HSPCs compared to the total endothelial cell population in the trunks, these results indicate that both driver/effector combinations are functional and achieve targeted overexpression of *aplnrb* in the respective tissues.

Fig R2 tissue specific *aplnr* overexpression

Quantitative RT-PCR results for *aplnr* in control embryos and upon vascular *aplnr* overexpression mediated by *Tg(fli1:GAL4FF); (TgUAS:aplnr)* (unpaired t-test, $p = 0.0104$) and upon hemogenic endothelium specific *aplnr* overexpression mediated by *Tg(gata2b:KALTA4); (TgUAS:aplnr)* (unpaired t-test, $p < 0.0001$). Methodical process: 3 biological replicates were conducted, per replicate and condition 50 tails were dissected and collected.

2. Fig 3: GFP in most cells labelled with arrowheads is very dim and hardly visible, thus it is unclear how the cells labelled with arrowheads were called positive. Authors should improve brightness / contrast.

We thank the reviewer for this valid point. We have increased brightness and contrast of Figure 3 to improve visibility.

3. Fig. 3: please clarify whether siblings to the *hsp70l:apl* transgenics were heat-shocked as well, which is the appropriate control for heat-shock experiments.

We thank the reviewer for this legitimate question. All embryos *Tg(hsp70l:apl)* and siblings were heat-shocked together. We included this information in the methods chapter.

4. Fig. EV7: did those drug treatments (Notch inhibitor, Shh activator, Wnt activator) produce other phenotypes? In the absence of such positive controls, the authors should not conclude that they do not affect *aplnr* expression.

We thank the Reviewer for this valuable suggestion. All chemical treatments were validated by established phenotypic readouts: RO4929097 reduced expression of the Notch reporter *Tg(EPV.Tp1-Mmu.Hbb:Venus-Mmu.Odc1)s940* (Ninov et al. 2012) and led to a strong reduction in *runx1* expression compared to controls (Fig. R1A–C); DMH1 disrupted caudal vein plexus formation (Nakajima et al. 2023) and similarly resulted in a strong reduction in *runx1* expression (Fig. R1A–C); LiCl treatment caused loss of eye structures (Van De Water et al. 2001) and led to a modest increase in *runx1* expression (Fig. R1D); Purmorphamine caused pooling of primitive blood in the posterior blood island (Wilkinson et al. 2009) but did not affect *runx1* expression (Fig. R1E).

Fig R1 *runx1* ISH after inhibitor/activator treatments

(A-E) WISH for *runx1* at 24 hpf upon chemical compound treatment. **(A)** Trunk region of DMSO treated control embryos. **(B)** Trunk region of Notch inhibitor (1 μ M RO4929097) treated embryos, treatment from 12 – 24 hpf. **(C)** Trunk region of BMP inhibitor (10 μ M DMH1) treated embryos, treatment from 12 - 40 hpf. **(D)** Trunk region of Wnt activator (0.15M LiCl) treated embryos, treatment from 12 - 24 hpf. **(E)** Trunk region of Shh activator (20 μ M Purmorphamine) treated embryos, treatment from 12 - 24 hpf.

5. Fig. 5H-J: can the authors give measures of reproducibility / variability for these data? Since the differences between wild-type and mutants is small and the data is pooled from just 4 fish each, in the absence of such data it is difficult to conclude that the observed differences are reproducible / statistically significant.

We thank the Reviewer for this constructive feedback. We repeated the whole kidney marrow (WKM) FACS analysis in three independent experiments, using four pooled WKMs per genotype per replicate. These analyses confirmed a significant increase in both myeloid and progenitor cell populations (Fig. 5H–J). Additionally, we have updated Figure 5J to present fold change over WT instead of raw percentages, thereby improving the clarity and comparability of the data.

Reviewer #3

General comment

*Eberlein et al. first identify that, over the course of 20 to 24 hpf, certain endothelial cells (marked by HRAS and KDR) in the ventral dorsal aorta lose apelin receptor expression. Given the location and developmental timing of these cells, the authors speculated that the cells may be contributing to the endothelial to hematopoietic transition (EHT). Upon sorting and qPCR of APLNRB-expressing and non-expressing cells, APLNRB non-expressing cells had higher levels of RUNX1 and lower levels of KDRL. The authors crossed the *ablnrb* reporter zebrafish with *itga2b* expressing cells, a marker of nascent hematopoietic stem and progenitor cells (HSPCs), and observed that apelin receptor non-expressing cells, also express the hematopoietic marker ITGA2B (CD41). Using time lapse imaging, the authors say that they*

can visualize the APLNR-negative cells budding into the dorsal aorta.

The authors then perform a series of experiments that eliminate expression of either apelin or the apelin receptor and examine the number of *itga2b*⁺ cells and/or *runx1* expression. The data suggests that elimination of apelin signalling increases the number of hematopoietic cells and *runx1* expression. In contrast, overexpression of *apln* does not decrease hematopoietic output, but overexpression of the *aplnrb* does, indicating that it is the lack of *aplnrb* expression on these cells that is supportive of the hematopoietic state. Finally, the knockout of apelin using CRISPRs increases the number of HPSCs in the VDA, with some cells co-expressing *gata2b* and *aplnrb*, indicating that, if apelin signalling were present, these cells would likely remain in an endothelial state.

Mechanistically, *aplnrb* knockout cells do not have higher proliferation, indicating that this increase in hematopoietic progenitors is not simply due to a differential proliferation phenotype. Further, they use small molecule inhibitors to show that inhibition of Notch, BMP4, Wnt or Sonic hedgehog activity does not impact apelin receptor expression. *Aplnr*b overexpression from *gata2b* or *runx1* promoters, show no difference in HSPC emergence, while knockout of *runx1* has no impact on apelin signalling. This evidence indicates that apelin signalling, or lack thereof, is important prior to the EHT, in the initial endothelial commitment stage. Overall, the authors show that modulation of apelin signalling early in development can alter HSPC emergence. Understanding the timing and the role of apelin is informed by the studies performed, but questions remain and interpretation is challenging.

Major comments:

1. The authors use of a short-lived fluorescent reporter Venus-PEST enables more specific quantification of the kinetics of this process, a strength of this paper. However, the timeline of this process is quite confusing. While they start to see downregulation of *aplnrb* at 22hpf, they also mention in the introduction that *gata2b*, the earliest marker of the EHT, can be detected at 20hpf. Thus, does this process happen after the onset of the EHT, following *gata2b* expression? Timelines representing the experimental frameworks and to summarize the findings at the end would help to increase comprehension in this area.

We thank the Reviewer for this valid question. The reference to *gata2b* expression in the introduction refers to endogenous mRNA detection by *in situ* hybridization, as described by Butko et al. (2015). In contrast, our analysis of *aplnrb* expression dynamics is based on the *aplnrb*:Venus-PEST reporter. While Venus-PEST is a destabilized fluorophore with a markedly reduced half-life, complete degradation is not instantaneous and may introduce a slight delay compared to endogenous *aplnrb* transcript dynamics. Therefore, we interpret the observed downregulation of the *aplnrb* reporter in hemogenic endothelial cells as occurring shortly after the initiation of *gata2b* expression. We added this information into the discussion.

2. Many of the images provided are hard to interpret and unconvincing as evidence for general conclusions. Please provide quantification to support all main conclusions discussed in the text. For instance, the conclusion that *aplnrb* downregulation is needed prior to the EHT is unfounded as there is no quantification of how many of the budding cells arise from HEC that have downregulated *aplnrb*. While the images suggest that there is budding occurring from *aplnrb*⁻ cells, it can be hard to interpret the images provided. Quantification such as the percentage of cells budding from *aplnrb*⁻ vs *aplnrb*⁺ cells would supplement these conclusions and give context to the requisite nature of this *aplnrb* downregulation.

We thank the reviewer for this excellent suggestion. We further analysed the nascent HSPCs to determine the proportion arising from *aplnrb* non-expressing HECs. We observed that almost all HSPCs (Mean = 99,1825 %, n = 23 embryos) in the VDA arise from *aplnrb* non-expressing HECs and included these findings into Figure 1 J.

Minor comments:

1. *The use of zebrafish genetic terminology, while required, is hard to follow. Improvement of readability would help readers understand the significance of this work.*

We thank the Reviewer for this helpful suggestion. To improve readability and facilitate cross-species comparison, we have added the corresponding mammalian gene names in brackets throughout the manuscript wherever applicable.

2. *There is very little information about apelin signalling mechanisms. While the authors show apelin receptor b and apelin expression, and that modulation of those components can alter the number of budding cells, they do not ever show active signalling within the cells still possessing the apelin receptor. Downstream target investigation from the apelin receptor would help to understand how the signalling alters the current thinking about the EHT requirements. Given that mechanism is not needed for the scope of this journal, adding to the introduction and discussion about what is known about apelin signalling would help to situate how the signalling of this process integrates with current knowledge of the EHT.*

We thank the Reviewer for this valid point. Currently, there are no established assays to directly measure endogenous GPCR activity or to visualize GPCR–G-protein coupling and downstream signaling events *in vivo*. While this limits our ability to directly demonstrate downstream signaling at the cellular level, we agree that additional context would strengthen the manuscript. As suggested, we have expanded the Introduction to include relevant information on Aplnr-associated G-protein coupling and known downstream signaling pathways.

3. *Please ensure the statistical tests used are the appropriate choices. For example, in the FACS of aplnr^b expressing cells, the authors used an unpaired t-test to compare the two conditions. In this reviewer's experience, these experiments are usually paired, with each sort representing a biological replicate. If this is the case, a paired t-test would be a more appropriate comparison.*

We thank the Reviewer for this helpful suggestion and did use a paired t-test as the appropriate statistical test for the analysis of aplnr^b-expressing and aplnr^b non-expressing FACS sorted cells. The corresponding panels in Figure 1 and Figure EV1 have been exchanged.

4. *The erythroid cells from apln mutants have much different side scatter profiles than those in wild type. Does this indicate a different phenotype of those erythroid cell type?*

We thank the Reviewer for this valid question. The observed shift in side scatter (SSC) profiles of erythroid cells in apln mutants may indeed reflect changes in internal complexity or granularity, potentially indicating a phenotypic difference. However, a detailed characterization of this alteration was not pursued, as it falls outside the scope of the current study.

5. *In Calvanese et al. (2022) "Mapping Human Hematopoietic Cells from Hemogenic Endothelium to Birth", there is full sequencing of the human EHT process. Confirming the expression of aplnr^b and apln throughout this process would help to solidify its relevance within the human system.*

We thank the Reviewer for this helpful suggestion. The data published by Calvanese and colleagues (Calvanese et al. 2022) supports our observations in the zebrafish. Analysis of the published datasets showed that in humans APLNR is enriched in the aortic endothelium (cluster 2), but absent from the HE and HSCs. Notably, APLNR is among the top downregulated genes in the HE compared with all other clusters. We have included this information in the Discussion section of the manuscript (p. 21, line 15 ff) to further highlight the conserved nature of Apelin receptor expression dynamics during EHT.

References

- Bonkhofer, Florian, Rossella Rispoli, Philip Pinheiro, Monika Krecsmarik, Janina Schneider-Swales, Ingrid Ho Ching Tsang, Marella de Bruijn, Rui Monteiro, Tessa Peterkin, and Roger Patient. 2019. "Blood Stem Cell-Forming Haemogenic Endothelium in Zebrafish Derives from Arterial Endothelium." *Nature Communications* 10(1):3577.
- Butko, Emerald, Martin Distel, Claire Pouget, Bart Weijts, Isao Kobayashi, Kevin Ng, Christian Mosimann, Fabienne E. Poulain, Adam McPherson, Chih-Wen Ni, David L. Stachura, Natasha Del Cid, Raquel Espín-Palazón, Nathan D. Lawson, Richard Dorsky, Wilson K. Clements, and David Traver. 2015. "Gata2 Is a Restricted Early Regulator of Hemogenic Endothelium in the Zebrafish Embryo." *Development* 142(6):1050–61.
- Calvanese, Vincenzo, Sandra Capellera-Garcia, Feiyang Ma, Iman Fares, Simone Liebscher, Elizabeth S. Ng, Sophia Ekstrand, Júlia Aguadé-Gorgorió, Anastasia Vavilina, Diane Lefaudeux, Brian Nadel, Jacky Y. Li, Yanling Wang, Lydia K. Lee, Reza Ardehali, M. Luisa Iruela-Arispe, Matteo Pellegrini, Ed G. Stanley, Andrew G. Elefanty, Katja Schenke-Layland, and Hanna K. A. Mikkola. 2022. "Mapping Human Haematopoietic Stem Cells from Haemogenic Endothelium to Birth." *Nature* 604(7906):534–40.
- Helker, Christian SM, Jean Eberlein, Kerstin Wilhelm, Toshiya Sugino, Julian Malchow, Annika Schuermann, Stefan Baumeister, Hyouk-Bum Kwon, Hans-Martin Maischein, Michael Potente, Wiebke Herzog, and Didier YR Stainier. 2020. "Apelin Signaling Drives Vascular Endothelial Cells towards a Pro-Angiogenic State." *ELife* 9:1–20.
- Herd, Lukas, Stefan Baumeister, Jeshma Ravindra, Jean Eberlein, and Christian S. M. Helker. 2025. "Apelin as a CNS-Specific Pathway for Fenestrated Capillary Formation in the Choroid Plexus." *Nature Communications* 16(1):7729.
- Kwon, Hyouk Bum, Shengpeng Wang, Christian S. M. Helker, S. Javad Rasouli, Hans Martin Maischein, Stefan Offermanns, Wiebke Herzog, and Didier Y. R. Stainier. 2016. "In Vivo Modulation of Endothelial Polarization by Apelin Receptor Signalling." *Nature Communications* 7(May):1–12.
- Lundin, Vanessa, Wade W. Sugden, Lindsay N. Theodore, Patricia M. Sousa, Areum Han, Stephanie Chou, Paul J. Wrighton, Andrew G. Cox, Donald E. Ingber, Wolfram Goessling, George Q. Daley, and Trista E. North. 2020. "YAP Regulates Hematopoietic Stem Cell Formation in Response to the Biomechanical Forces of Blood Flow." *Developmental Cell* 52(4):446-460.e5.
- Mahony, Christopher B. and Rui Monteiro. 2024. "Protocol for the Analysis of Hematopoietic Lineages in the Whole Kidney Marrow of Adult Zebrafish." *STAR Protocols* 5(1):102810.
- Malchow, Julian, Jean Eberlein, Wei Li, Benjamin M. Hogan, Kazuhide S. Okuda, and Christian S. M. Helker. 2024. "Neural Progenitor-Derived Apelin Controls Tip Cell Behavior and Vascular Patterning." *Science Advances* 10(27).
- Nakajima, Hiroyuki, Hiroyuki Ishikawa, Takuya Yamamoto, Ayano Chiba, Hajime Fukui, Keisuke Sako, Moe Fukumoto, Kenny Mattonet, Hyouk Bum Kwon, Subhra P. Hui, Gergana D. Dobrova, Kazu Kikuchi, Christian S. M. Helker, Didier Y. R. Stainier, and Naoki Mochizuki. 2023. "Endoderm-Derived Islet1-Expressing Cells Differentiate into Endothelial Cells to Function as the Vascular HSPC Niche in Zebrafish." *Developmental Cell* 58(3):224-238.e7.
- Ninov, Nikolay, Maxim Borius, and Didier Y. R. Stainier. 2012. "Different Levels of Notch Signaling Regulate Quiescence, Renewal and Differentiation in Pancreatic Endocrine Progenitors." *Development* 139(9):1557–67.
- Qi, Jialing, Annegret Rittershaus, Rashmi Priya, Shivani Mansingh, Didier Y. R. Stainier, and

- Christian S. M. Helker. 2022. "Apelin Signaling Dependent Endocardial Protrusions Promote Cardiac Trabeculation in Zebrafish." *ELife* 11:1–20.
- Soto, Rebecca A., Mohamad Ali T. Najia, Mariam Hachimi, Jenna M. Frame, Gabriel A. Yette, Edroaldo Lummertz da Rocha, Kryn Stankunas, George Q. Daley, and Trista E. North. 2021. "Sequential Regulation of Hemogenic Fate and Hematopoietic Stem and Progenitor Cell Formation from Arterial Endothelium by *Ezh1/2*." *Stem Cell Reports* 16(7):1718–34.
- Van De Water, S., M. Van De Wetering, J. Joore, J. Esseling, R. Bink, H. Clevers, and D. Zivkovic. 2001. "Ectopic Wnt Signal Determines the Eyeless Phenotype of Zebrafish Masterblind Mutant." *Development* 128(20):3877–88.
- Wilkinson, Robert N., Claire Pouget, Martin Gering, Angela J. Russell, Stephen G. Davies, David Kimelman, and Roger Patient. 2009. "Hedgehog and Bmp Polarize Hematopoietic Stem Cell Emergence in the Zebrafish Dorsal Aorta." *Developmental Cell* 16(6):909–16.

Dear Dr. Helker

Thank you for the submission of your revised manuscript to EMBO reports. We have meanwhile received the full set of referee reports that is copied below.

As you will see, all referees are very positive about the study and request only minor changes to clarify text and figures.

From the editorial side, there are also a few things that we need before we can proceed with the official acceptance of your study.

Please provide a point-by-point response to the remaining referee concerns and to the editorial points listed below to speed up the final checks.

1) Please remove the figures from the manuscript text file. The main and EV figure legends should both be part of the main manuscript text file and go to the end of the manuscript.

2) The manuscript sections should be in the following order: Title page - Abstract & Keywords - Introduction - Results - Discussion - Methods - Data Availability - Acknowledgments - Disclosure Statement & Competing Interests - References - Figure Legends - (Main Tables with legends if applicable) - Expanded View Figure Legends.

3) Please provide up to 5 keywords on the title page.

4) Please remove the "Data and materials availability" section on page 27. It is sufficient to have the Data Availability section at the end of materials and methods.

5) Please rename the Competing Interest statement to Disclosure and Competing Interests Statement.

6) Regarding the Author Contributions, we now use CRediT to specify the contributions of each author in the journal submission system. Therefore, please remove the Author Contributions from the manuscript file and make sure that the author contributions in our online manuscript tracking system are correct and up-to-date. The information you specified in the system will be automatically retrieved and typeset into the article. You can enter additional information in the free text box provided, if you wish. See also our guide to authors <https://www.embopress.org/page/journal/14693178/authorguide#authorshipguidelines>.

7) Please remove the "notes" from the "References and Notes" header.

8) All information on funding needs to be part of the Acknowledgments section. The "Funding" header is not needed. In addition, all information on funding must be specified in the online manuscript tracking system and the information in the manuscript and in the system must be congruent.

9) Please add callouts in the text for the following panels/figures: Figure 6HI, Figure EV2.

10) Please download and fill our Reagents and Tools Table template (.docx), which you can find in our author guidelines: <https://www.embopress.org/page/journal/14693178/authorguide#structuredmethods>.

11) Table 1-4 could e.g., also go to the Reagents and Tools table.

12) Could you please add the reference number for the approval of animal experiments in the methods section (see Author Checklist, line 95).

13) Please address the following points in the figure legends:

a) Please note that information related to n is missing in the legends of figures EV1 H

b) Please note that the error bars are not defined in the legends of figures 1F, G, J; 2C, 3C, F, I; 4C, F, G; 5C, J; 6G, EV1 H, EV2, EV3 E, F; EV4 A, B

14) Source data: please upload the source data as one zipped folder per figure. While you have provided all quantification data as .xls files, we would also require source data for all microscopy images. These could either be uploaded as part of the source data folders or alternatively, if large, be uploaded to public repositories such as FigShare or Biostudies.

15) Finally, EMBO Reports papers are accompanied online by
A) a short (1-2 sentences) summary of the findings and their significance,
B) 2-3 bullet points highlighting key results and
C) a schematic summary figure that provides a sketch of the major findings (not a data image).
Please provide the summary figure as a separate file in PNG or JPG format at a size of 550x300-600 pixels (width x height).
Please note that the size is rather small and that text needs to be readable at the final size. Please send us this information
along with the revised manuscript.

With kind regards,

=====

Referee #1:

I thank the authors for addressing most of my comments and those of the other reviewers; overall the manuscript has improved and I'm happy to recommend publication of this elegant study.

There is an issue, however, that I would still like addressed:

- response to major comment #2 - I agree with the authors that the 54hpf data is a good timepoint to count EMPs at, and they clearly show no difference between genotypes. However, at 3dpf, cd41-gfp/gata1/dsred cells could be EMPs or HSPC-derived erythroid progenitors; thus, presenting the 3dpf count of these cells at 72hpf as an EMP count alone is not accurate and might induce others in the field in error. I would suggest to amend the text to reflect this, or remove the 3dpf data as it doesn't materially affect the author's conclusions.

Referee #2:

The authors have dealt with all of my concerns in an adequate manner, and have thus even further strengthened an - in my opinion - already excellent manuscript. I think that the presented work will be of great interest to readers of EMBO Rep and highly recommend publication.

Referee #3:

The author's have done a good job addressing the reviewer comments.

Martina Rembold, Ph.D.
Senior Editor
On behalf of EMBO reports

November 04, 2025

Dear Martina,

Please find attached our revised manuscript entitled “**Apelin signaling acts as a molecular switch between endothelial and hematopoietic stem cell fates**” for consideration as a Research Article in *EMBO Reports*. We thank you for the positive feedback and for giving us the opportunity to resubmit.

Please find below our point-by-point response to the editorial requests and the reviewers' comments.

Best regards,

Christian Helker

Editorial requests

1) Please remove the figures from the manuscript text file. The main and EV figure legends should both be part of the main manuscript text file and go to the end of the manuscript.

Done, as requested.

2) The manuscript sections should be in the following order: Title page - Abstract & Keywords - Introduction - Results - Discussion - Methods - Data Availability - Acknowledgments - Disclosure Statement & Competing Interests - References - Figure Legends - (Main Tables with legends if applicable) - Expanded View Figure Legends.

Done, as requested.

3) Please provide up to 5 keywords on the title page.

Done, as requested.

4) Please remove the "Data and materials availability" section on page 27. It is sufficient to have the Data Availability section at the end of materials and methods.

Done, as requested.

5) Please rename the Competing Interest statement to Disclosure and Competing Interests Statement.

Done, as requested.

6) Regarding the Author Contributions, we now use CRediT to specify the contributions of each author in the journal submission system. Therefore, please remove the Author Contributions from the manuscript file and make sure that the author contributions in our online manuscript tracking system are correct and up-to-date. The information you specified in the system will be automatically retrieved and typeset into the article. You can enter additional information in the free text box provided, if you wish. See also our guide to authors <https://www.embopress.org/page/journal/14693178/authorguide#authorshipguidelines>.

Done, as requested.

7) Please remove the "notes" from the "References and Notes" header.

Done, as requested.

8) All information on funding needs to be part of the Acknowledgments section. The "Funding" header is not needed. In addition, all information on funding must be specified in the online manuscript tracking system and the information in the manuscript and in the system must be congruent.

Done, as requested.

9) Please add callouts in the text for the following panels/figures: Figure 6HI, Figure EV2.

Figure 6 H/I does not exist. Callouts for Figure EV2 has been added to the text.

10) Please download and fill our Reagents and Tools Table template (.docx), which you can find in our author guidelines:

An example of a Method paper with Structured Methods can be found here:
<https://www.embopress.org/doi/10.15252/msb.20178071>.

Done, as requested.

11) Table 1-4 could e.g., also go to the Reagents and Tools table.

Tables 1-4 was removed from the manuscript and added to the Reagents and tools table.

12) Could you please add the reference number for the approval of animal experiments in the methods section (see Author Checklist, line 95).

Done, as requested.

13) Please address the following points in the figure legends:

- a) Please note that information related to n is missing in the legends of figures EV1 H
- b) Please note that the error bars are not defined in the legends of figures 1F, G, J; 2C, 3C, F, I; 4C, F, G; 5C, J; 6G, EV1 H, EV2, EV3 E, F; EV4 A, B

Done, as requested.

14) Source data: please upload the source data as one zipped folder per figure. While you have provided all quantification data as .xls files, we would also require source data for all microscopy images. These could either be uploaded as part of the source data folders or alternatively, if large, be uploaded to public repositories such as FigShare or Biostudies.

Done, source data uploaded to BioImage Archive.

15) Finally, EMBO Reports papers are accompanied online by

- A) a short (1-2 sentences) summary of the findings and their significance,
- B) 2-3 bullet points highlighting key results and
- C) a schematic summary figure that provides a sketch of the major findings (not a data image).

Please provide the summary figure as a separate file in PNG or JPG format at a size of 550x300-600 pixels (width x height). Please note that the size is rather small and that text needs to be readable at the final size. Please send us this information along with the revised manuscript.

Done, Synopsis text and summary figure are uploaded in the online submission system.

Referee comments

Referee #1:

I thank the authors for addressing most of my comments and those of the other reviewers; overall the manuscript has improved and I'm happy to recommend publication of this elegant study.

There is an issue, however, that I would still like addressed:

- response to major comment #2 - I agree with the authors that the 54hpf data is a good timepoint to count EMPs at, and they clearly show no difference between genotypes. However, at 3dpf, cd41-gfp/gata1/dsred cells could be EMPs or HSPC-derived erythroid progenitors; thus, presenting the 3dpf count of these cells at 72hpf as an EMP count alone is not accurate and might induce others in the field in error. I would suggest to amend the text to reflect this, or remove the 3dpf data as it doesn't materially affect the author's conclusions.

We thank the reviewer for this valid point and excluded the data regarding EMPs at 72 hpf from the manuscript. The updated Figure EV4 has been uploaded.

Referee #2:

The authors have dealt with all of my concerns in an adequate manner, and have thus even further strengthened an - in my opinion - already excellent manuscript. I think that the presented work will be of great interest to readers of EMBO Rep and highly recommend publication.

We thank the reviewer for the kind words and the positive feedback.

Referee #3:

The author's have done a good job addressing the reviewer comments.

We thank the reviewer for the positive feedback.

Dr. Christian Helker
Marburg University
Department of Biology, Animal Cell Biology
Marburg
Germany

Dear Dr. Helker,

I am very pleased to accept your manuscript for publication in the next available issue of EMBO reports. Thank you for your contribution to our journal.

Kind regards,
